# Influence of ozone and humidity on PTR-MS and GC-MS VOC measurements with and without $Na_2S_2O_3$ ozone scrubber

Lisa Ernle[1], Monika Akima Ringsdorf[1], and Jonathan Williams[1]

[1]Max Planck Institute for Chemistry, Hahn-Meitner-Weg 1, 55128 Mainz, Germany

**Correspondence:** jonathan.williams@mpic.de

**Abstract.** The measurement of volatile organic compounds (VOCs) can be influenced by ozone ($O_3$), resulting in sampling artifacts that corrupt the data obtained. Published literature reports both, positive (false enhancements of signal) and negative (loss of signal) interference in VOC data due to ozonolysis occurring in the sample gas. To assure good data quality it is essential to be aware of such interfering processes, characterize them and try to minimize the impact with a suitable sampling
setup. Here we present results from experiments with a sodium thiosulfate ozone scrubber ($Na_2S_2O_3$), which is a cost effective and easily applied option for $O_3$ scavenging during gas phase sampling. Simultaneous measurement of selected organic trace gases using gas chromatography- and proton transfer reaction-mass spectrometry was performed at different ozone levels (0-1 ppm) and different relative humidities (0-80 %). In this way both tropospheric and stratospheric conditions were examined. The measured data show that several carbonyl compounds including acetaldehyde, acetone and propanal show artificial signal
enhancement when ozone is present at higher concentrations (>150 ppb) in dry air, while analytes with double bonds like isoprene (measured with GC-MS) and terpenes show lower signals due to reaction with ozone. Both effects can be eliminated or in the case of sesquiterpenes substantially reduced by using $Na_2S_2O_3$ impregnated quartz filters in the inlet line. With the chosen scrubbing material, relative humidity (RH) substantially improves the scrubbing efficiency. Under surface conditions between 50-80 % RH, the filter allows accurate measurement of all species examined.

## 1 Introduction

Volatile organic compounds (VOCs) are trace atmospheric constituents usually present at mixing ratios of parts per billion (ppb) or lower in the earth's atmosphere. Nevertheless, they can have a considerable impact on the global air chemistry, climate and influence the health of living organisms on the earth's surface (Crutzen and Lelieveld, 2001; Williams, 2004). Organic trace gases can act as greenhouse gases, contribute to particle formation and take part in photochemical oxidation processes
that influence ozone. Additionally many VOCs are also considered to be contaminants of the indoor environment where human exposure to such chemicals can be high (Weschler and Carslaw, 2018). VOC sources can be of natural (plants, phytoplankton, volcanoes, etc.) or anthropogenic origin (e.g. fossil fuel combustion, agriculture, industry) (Koppmann, 2008; McDonald et al., 2018; Weschler and Shields, 1997). To understand the chemical reactions and processes in outdoor as well as indoor environments, it is essential to accurately quantify the VOCs in the air.

Common analytical techniques for sampling organic trace gases are proton-transfer-reaction mass spectrometry (PTR-MS) and

gas chromatography-mass spectrometry (GC-MS). With these measurement techniques a wide range of volatile organic compounds can be measured including aliphatic and aromatic hydrocarbons, oxygenated and halogen containing species (Koppmann, 2008; Warneck and Williams, 2012). However, research has shown that due to the reactivity of some analytes to ozone, measurements can be rendered inaccurate, as already reported by Helmig (1997) more than twenty years ago.

Ozone can influence VOC measurements either due to reaction with the target analytes during sampling, which particularly affects techniques with pre-concentration steps prior to analysis (Helmig, 1997; Koppmann et al., 1995; Pollmann et al., 2005), or by generating sampling artifacts in the inlet of an online instrument. Northway et al. and Apel et al. reported for example increased mixing ratios for acetaldehyde in their systems for measurements in the lower stratosphere where ozone levels are high and humidity is low. Ozone and water are omnipresent in the troposphere with mixing ratios between 10 and 200 ppb

(ozone) and humidities (10-100 %). Stratospheric $O_3$ is essential for life on planet earth as it absorbs high energy solar UV radiation. It is formed in the stratosphere through the photolysis of oxygen, which generates mixing ratios between 1-10 ppm. While such processes generate the protective ozone layer in the stratosphere, at ground level this oxidative gas is considered to be a pollutant as it is detrimental to the human respiratory tract and damages plants (Pandis and Seinfeld, 2006). Being present in both the troposphere and stratosphere, albeit at different concentrations, ozone can potentially affect VOC measurements

made at the ground and from high flying aircraft. The two instruments examined in this study are regularly installed on an aircraft capable of reaching ca. 15 km which at mid-latitudes gives access to the lower stratosphere. Also in indoor environments where ozone is typically 3-5 times less than outside ambient levels, it may affect the measurement fidelity.

Sodium thiosulfate ($Na_2S_2O_3$) has been reported to have a good ozone scrubbing efficiency for VOC measurements (Lehmpuhl and Birks, 1996; Pollmann et al., 2005) and was therefore chosen as the best test material for $O_3$ removal in the experiments

described here. Like ozone, humidity varies strongly between the dry stratosphere and much more humid conditions of the earth's surface. Humidity is considered an important variable here as it can strongly influence chemistry occurring at surfaces. Measurement conditions were therefore examined that reflect the conditions likely to be met in these two environments. The aim of this study was to investigate the influence of different ozone levels and different relative humidity (RH) on two specific VOC measurement instruments, namely the fast GC-MS "SOFIA" described by Bourtsoukidis et al. (2017) and a PTR-ToF-

MS described in Wang et al. (2022). We report on the effect of using the sodium thiosulfate impregnated quartz filters for multiple VOC species including carbonyls, alcohols and non-methane hydrocarbons. Additionally, the lifetime of the scrubber with respect to ozone exposure was determined under tropospheric and stratospheric conditions. This is essential to assure that the scrubber is working correctly when applied in the field, to determine optimum exchange times, and to avoid unnecessary exchange and waste. These results will define operational expectations of using $Na_2S_2O_3$ filter in field conditions, to

improve organic trace gas measurement techniques, assure good data quality for smaller VOCs and therefore improve the data comparability between different studies.

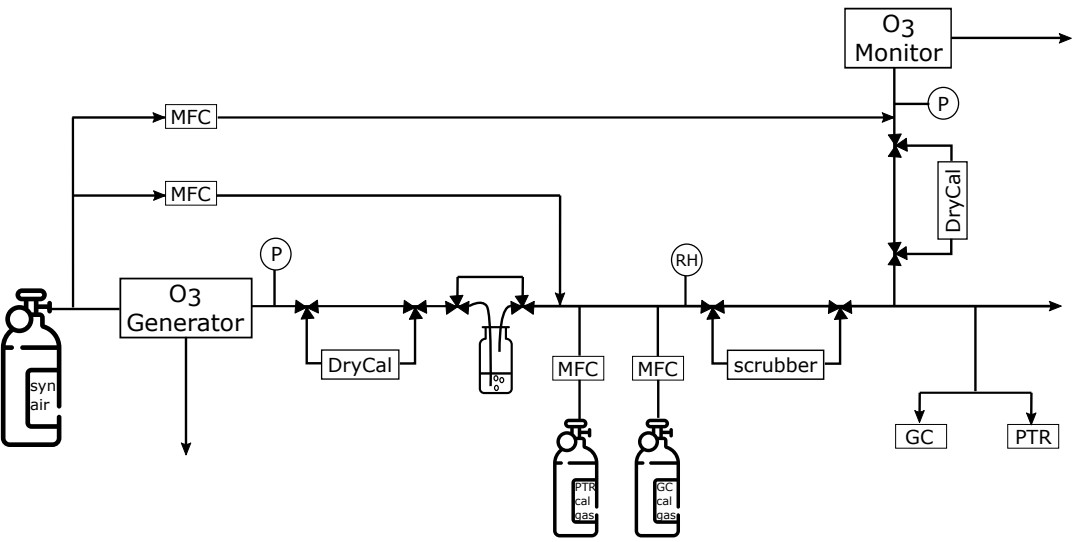

**Figure 1.** Experimental setup, abbreviations stand for: MFC - mass flow controller; RH - relative humidity sensor; P - pressure gauge; DryCal - flow meter.

## 2 Materials and methods

The influence of ozone on measurements from an online VOC instrument (PTR-ToF-MS 8000, Ionicon Analytik, Austria) and a custom built quasi-online fast gas chromatograph-mass spectrometer (GC-MS) (Bourtsoukidis et al., 2017) which collects,
concentrates and measures within 3 minutes was investigated. Additionally, the effect of an implemented sodium thiosulfate ozone scrubber on those systems as well as the scrubber lifetime was examined at different relative humidity. A list with the measured species is shown in Table 2. It includes saturated and unsaturated halocarbons, non-methane hydrocarbons (NMHCs), small oxygenated VOCs (OVOCs), siloxanes and some nitrogen and sulfur containing molecules. Some species could be detected with both instruments simultaneously while other species could only be measured by one. The majority of the tubing
used in this study was FEP (fluorinated ethylene propylene) Teflon which was found by Deming et al. to perform well in a comparison of inlet materials (including polyetheretherketone (PEEK) and stainless steel), as adsorption on FEP was found to be independent of humidity, concentration and functionality. The tubing was not new, but used previously for airborne measurements aboard a research aircraft. It was flushed with synthetic air for at least one hour prior to the experiments performed. When considering the ozone in the instrument inlet, one could consider passivating the inlet surfaces prior to measurement
by the introduction of high (500 ppb) ozone mixing ratios. Northway et al. tested this possibility and noted a passivation that disappeared during further field measurements. As this will in effect generate a shifting background to the subsequent measurements, and as 6 hour flushing is impractical prior to flight measurements we chose not to follow this procedure.

**Table 1.** Conditions of the different experiments performed in this study.

| Condition | $O_3$ levels / ppb | Calgas MR / ppb | RH / % |
|---|---|---|---|
| Effect of $O_3$ on VOCs | 0, 50, 1000 | 0, 0.5, 1, 2, 4 | 0 |
| Effect of $O_3$ on tubing | 0, 25, 50, 100, 150, 400, 750, 1000 | 0, 0.5 | 0 |
| Effect of RH on VOCs / scrubber | 0, 50, 150 | 0, 0.5, 2 | 0, 50, 80 |
| Scrubber endurance | 50, 150, 1000 | 0 | 0, 80 |

## 2.1 Experimental setup

The experimental setup includes both PTR-ToF-MS and fast GC-MS as shown in Figure 1. Synthetic air from a gas cylinder
(synthetic air, hydrocarbon free, T50, Westfalen, Germany) was connected to the ozone generator which was set to generate $O_3$
at levels between 25 and 1000 ppb. This corresponds to tropospheric and lower stratospheric levels, similar to those encountered
by the HALO aircraft on which the instruments are certified to fly. Thereafter, the air stream was led through ultra-pure water
for humidification. Ozone as a non-polar molecule has a very low solubility in water and will therefore not be lost during the
humidification process. Two different calibration gas cylinders (40L, Apel Riemer, USA) were connected through a junction
(T-piece) to the sampling line. These cylinders contained a gravimetrically prepared mixture of VOCs at known mixing ratios,
which could be added to the system air. The VOC enriched synthetic air could then be directed through the ozone scrubber
or directly to the two mass spectrometers, which both drew circa 200 sccm air from the sampling line each. At the end of the
Teflon tubing inlet line the ozone monitor was connected to measure the $O_3$ concentration after the scrubber. Several exhaust
lines and pressure gauges were installed in the setup to ensure a system pressure close to ambient pressure (ca. 1000 hPa). This
is required for the operation of the ozone instruments. To make sure that the monitor was always operated within the required
flow range, a dilution flow could be added shortly before the instrument. Gas flows were checked regularly with a flow meter
(DryCal, Mesalabs, USA) after the ozone generator and monitor. All the lines were made of FEP and heated to approximately
45 °C to avoid condensation in the tubing. This simulates conditions on the research aircraft aboard which the measurement
devices have been installed. The total tubing length between ozone generator and scrubber was approximately 2 m, the outer
diameter was 1/4", again equivalent to the aircraft set-up.

 During the ozone experiment series various tests were performed (cf. Table 1): VOC mixtures were measured at several levels,
adding 0, 50 and 1000 ppb of ozone to investigate the measurement performance at ambient background ozone levels on the
ground and in the lower stratosphere. Additionally, one VOC level was measured at seven different ozone levels up to 1000 ppb
(0, 25, 50, 150, 400, 750, 1000 ppb). The tests also included experiments without added VOCs to see if any of the analytes are
produced in the pre-used inlet line. Note that during the very first experiment performed, flows were measured every time the
VOC level was adjusted. It turned out that some compounds were emitted from the flow meter resulting in elevated terpene
masses. When switching to a new calibration gas level as well as in the first hour of the experiment, there were spikes in the
VOC signal. These were judged to be mechanical flow related anomalies and therefore removed to assure better visibility of

the mixing ratio in the plots. To investigate the influence of sodium thiosulfate impregnated quartz filters as an ozone scrubber, VOC levels at 0.5 and 2 ppb were measured at different ozone levels (0, 50 and 150 ppb) and different relative humidities (0 %, 50 % and 80 % RH) with and without scrubber in the flow path. For scrubber endurance tests, another exhaust line was installed after the ozone generator to reduce the flow through the sodium thiosulfate impregnated quartz filter in order to simulate field conditions. The flow through the filter was set to 200-600 sccm to replicate either typical field measurement conditions or the experiments focusing on the influence on the VOC measurements. For those experiments the flow through the scrubber was higher to provide enough air for both of the instruments. Before each single longevity experiment the $Na_2S_2O_3$ impregnated filter was exchanged and left in the flow path until an abrupt increase in ozone concentration was detected with the $O_3$ monitor. This rise marks the end of the scrubber lifetime. Further details on the preparation of the filter scrubber can be found in section 2.3. Scrubber performance was tested under three different ozone concentrations: 50, 150 and 1000 ppb at 0 % RH. These are the same ozone levels used for the experiment focused on the effect of $O_3$ on the VOC measurements, which correspond to ambient ground and lower stratospheric ozone levels. Additionally, the influence of 80 % relative humidity on the scrubber lifetime was tested. This RH level was chosen as an extreme to see whether or not it changes the scrubber performance. Relative humidity was measured with a humidity sensor that includes a temperature sensor (MSR145, MSR, Switzerland; indicated with "RH" in Figure 1).

## 2.2 Instrumentation

### 2.2.1 Ozone instruments

An ozone generator as well as an ozone monitor (49iQPS and 49iQ, both Thermo Fisher Scientific, USA) were installed in the experimental setup. Each instrument's supply air stream is divided into a reference and a sample gas channel. In the generator the sample gas flows through an ozonator, while the reference gas of the monitor flows through a scrubber to eliminate existing ambient ozone. Both instruments use a spectroscopic approach to determine the mixing ratio: as the $O_3$ molecule absorbs UV light of 254 nm the difference of light intensity of this wavelength in both channels is used to calculate the ozone concentration in the sample stream based on Beer's Law (ThermoFisherScientific, 2020a, b).

### 2.2.2 PTR-MS

An Ionicon PTR-ToF-MS with a drift tube pressure of 2.2 hPa, drift temperature 60 °C and E/N 137 Td was operated with $H_3O^+$ as primary ions. It is a soft ionization technique and therefore causes little fragmentation of the analytes during the detection process. This is the case for most analytes in this study. However, some species (e.g. terpenes, siloxanes) do fragment during ionization (Pagonis et al., 2019). Fragments can impact the measurement of target species such as isoprene if they have exactly the same mass. Identification of the analytes was performed using the exact mass of the most abundant fragment, usually the protonated molecular mass, which does not exclude simultaneous measurement of isomeric compounds. The mass range of the system was 0-500 amu and the mass resolution approximately 3500. The PTR used a FEP inlet tubing (OD 1/4" (0.635 cm), inner diameter (ID) 1/8" (0.3175 cm)) with an inlet flow of 200 sccm. The distance between ozone scrubber and

PTR was 1.85 m, resulting in an inlet residence time $t_{res}$ of ca. 4 s. In order to regulate the pressure in the drift tube during flight measurements, the sample air passes and adjustable O-ring (fluorinated propylene monomer (FPM) or nitrile butadiene rubber (NBR), $t_{res} \leq 30$ ms). The influence of the O-ring on VOC measurements was found to be zero without $O_3$ present, but has not been tested separately under ozone exposure. Inside the instrument, a 1 m line (ID 0.1 cm) made of polyetheretherketone (PEEK) is used (ca. 70 sccm, $t_{res} \leq 1$ s, depending on the flow rate). limits of detection (LOD) were <0.05 ppb, with a total uncertainty of 15-20 %. Measured species included alkenes, siloxanes and OVOCs. For measurements at RH>0 a humid calibration was applied: The calibration was performed at the same relative humidity as the corresponding experiment.

### 2.2.3 GC-MS

The fast GC-MS system has been described in detail by Bourtsoukidis et al. (2017). In this study it was used to measure halocarbons, small NMHCs and OVOCS as well as some sulfur containing compounds and small organic nitrates. Due to its chromatographic column it is capable of separating isomeric compounds prior to detection (e.g. acetone and propanal). The custom-built instrument uses a cryogenic three step pre-concentration to collect air samples, followed by gas chromatographic separation in a custom-built oven and detection with a quadrupole mass spectrometer which was operated in selected ion monitoring mode (SIM). With a time resolution of 3 min it is currently not possible to measure high molecular mass compounds (e.g. sesquiterpenes) as those would need more time to elute from the GC-column. Bromoform is the largest analyte detected with the currently applied method. The system's inlet flow was 200 sccm, tubing length between GC inlet and ozone scrubber 2 m (OD 1/4" (0.635 cm), ID 1/8" (0.3175 cm)), which results in an inlet residence time of ca. 5 s. Inside the system, the sample air is exposed to silicosteel tubing (OD 1/16" (0.1588 cm), ID 0.02" (0.0508 cm), 40 sccm, $t_{res} < 1$ s) and stainless-steel surfaces in the traps ($t_{res}$ 1.5 min). LODs were typically <0.03 ppb (acetaldehyde, acetone and acrolein <0.2 ppb) and the total measurement uncertainty approximately 10 %.

### 2.3 Ozone scrubbing

Various materials have been tested to eliminate interferences from ozone on VOC measurements. Helmig (1997) compiled an overview of widely used $O_3$ scrubbing techniques for the sampling of atmospheric organic compounds. Several groups have reported satisfactory results of sodium thiosulfate as an ozone scavenger for VOC analysis (Helmig, 1997; Lehmpuhl and Birks, 1996; Pollmann et al., 2005; Strömvall and Petersson, 1992). In this study the scrubbers were prepared by soaking quartz fiber filters (37 mm, GE Healthcare Life Sciences, USA) in a 10 % (w/w) aqueous solution for 1 h followed by drying under a nitrogen flow of approximately 100 sccm at room temperature. This quartz filter was placed under a 47 mm PTFE-filter (Sartorius, Germany) in a Teflon filter holder. The smaller quartz filter was selected to avoid leaks at the filter holder (ID 47 mm, Reichelt Chemie Technik, Germany) previously caused due to the thickness of the quartz filter. The volume of the filter housing is ca. 55 mL, resulting in a residence time of ca. 6 s with a flow rate of ∼600 sccm.

## 2.4 Scrubber lifetime calculation

The time that the scrubber remains effective at removing ozone, here termed the scrubber lifetime is important information for field measurement practitioners. In order to improve data quality and keep cost and work load low, the ozone scrubbers need to be exchanged before their efficiency is compromised, while still using them as long as possible. Assuming that the scrubber lifetime is a function of ozone mixing ratio and flow, the data from the scrubber lifetime experiment was plotted and fitted with a power function. Additionally the influence of relative humidity on the scrubber lifetime was tested.

## 2.5 Potential effects causing interference

VOC measurements performed by the PTR-ToF-MS and the fast GC-MS may in the presence of ozone, suffer interference through various effects. Surface reactions on the inner walls of the tubing can lead to ozonolysis of compounds previously absorbed on the FEP inlet tubing. The ozonolysis of alkenes, which are either present on the tubing surface or in the gas phase (sample air) can lead to production of carbonyl compounds which cause positive artifacts on the carbonyl masses. Another potential source of interference is fragmentation during the ionization process in the PTR-MS. Several groups reported for example fragments on PTR *m/z* 69.07 from C5-C10 aldehydes (Buhr et al., 2002; Ruzsanyi et al., 2013; Wang et al., 2022). The instrument-internal fragmentation process itself is independent of ozone, but the presence of the aldehyde species in the sample air is likely to be caused by the release of those species from the sample line surface due to ozonolysis reaction. Not only the PTR, but also the GC-MS can suffer interference caused by ozone inside the instrument. It has been reported previously, that $O_3$ induced emission from rotor material of multiposition valves can lead to positive artifacts when measuring C2-C4 aldehydes (Apel et al., 2003).

## 3 Results and discussion

### 3.1 Influence of ozone on VOC measurements

For most of the analytes, mainly saturated NMHCs and halocarbons, no ozone interference of the measurement through reactive loss due to ozone was expected. This is because such species do not contain a double bond with which ozone can react, nor do they contain oxygen atoms so are unlikely to be produced by surface oxidation processes. However, due to the different molecular structures and physical properties of some analytes, it was potentially possible to obtain negative interference for some unsaturated species like isoprene and terpenes as well as artificial signal enhancement on the aldehyde masses. Table 2 shows whether ozone had an effect on the measured mixing ratio of the VOCs.

### 3.1.1 No effect on VOC measurements

In accordance with the expectations no interference from reactions with ozone were observed for most of the measured species, namely saturated and unsaturated halocarbons, alkanes, aromatics, nitriles, methyl tert butyl ether (MTBE), ethanol, hydroxy-acetone, methyl ethyl ketone (MEK), isopropyl nitrate, the two sulfur containing species carbon disulfide and dimethyl sulfide

**Table 2.** Measured species and effect of ozone. X: no effect; ↓: negatvive interference; ↑: positive interference.

| | Analytes and effect of O$_3$ on measured MR | | | | |
|---|---|---|---|---|---|
| X | Dichlorodifluoromethane | X | Trichloroethene | X | Hydroxyacetone |
| X | Trichlorofluoromethane (CFC-11) | X | Chlorobenzene | X | Isopropyl nitrate |
| X | Tetrachloromethane | X | Benzene | X | Acetonitrile |
| X | 1,1,2-Trichloro-1,2,2-Trifluoroethane (CFC-113) | X | m-Xylene | X | Acrylonitrile |
| X | Bromomethane | X | 1,2,4-Trimethylbenzene | X | Carbon disulfide |
| X | Bromoform | X | 3-Methylfuran | X | DMS |
| X | Chloromethane | X | 2-methyl-3-buten-1-ol | X | D3 |
| X | Dichloromethane | ↑ | Acetaldehyde | X | D4 |
| X | Chloroform | ↑ | Propanal | X | D5 |
| X | Iodomethane | ↑ | Acetone | X | n-butane, i-butane |
| X | Bromodichloromethane | ↑ | Butanal | X | Propene |
| X | 1,1,1-Trichloroethane | X | MEK | ↓ | Isoprene |
| X | 1,1,2-Trichloroethane | X | MTBE | ↓ | Monoterpenes |
| X | Vinylchloride | X | Acrolein | ↓ | Sesquiterpenes |
| X | Tetrachloroethene | ↓ | Methacrolein | | |

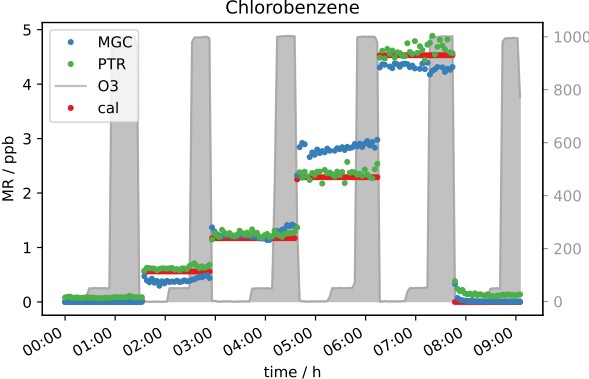

**Figure 2.** Effect of Ozone (0, 50, 1000 ppb) on GC- and PTR-MS measurements of chlorobenzene under different standard gas levels (0-4 ppb).

(DMS) as well as siloxanes. As an example of the VOCs where ozone did not influence the measurements Figure 2 shows the mixing ratios of chlorobenzene applied and measured by GC- and PTR-MS as well as the ozone mixing ratio. In this experiment no ozone scrubber was applied. While the instruments observe slightly different mixing ratios owing to differences in sensitivity, the relative change in chlorobenzene mixing ratios remain unchanged when ozone is present.

### 3.1.2 Ozone causing positive interference

Based on current literature, some carbonyl compounds such as aldehydes and ketones do have the potential to show higher mixing ratios when ozone is present. Northway et al. and Apel et al. observed positive artifacts under ozone presence for acetaldehyde. Additionally, Apel et al. observed artifact formation for propanal, acetone and butanal in their fast GC-MS system, which were emitted by parts of their system when ozone was present. Lehmpuhl and Birks found positive interference also for larger carbonyl compounds. Acetaldehyde measurements in this study under dry conditions agree with the results reported by Northway et al., Apel et al. and others. The VOC signal increases with ozone concentration. Both, the PTR- and GC-MS measured higher acetaldehyde mixing ratios when $O_3$ was above 150 ppb (see Figure 3). This indicates that the interference is not instrument specific but more likely a function of the common inlet tubing exposure to ozone. Note that the inlet lengths to GC and PTR were roughly the same. The higher enhancement of the GC acetaldehyde could be due to emission of oxidation products from the material of multiposition valves as described by Apel et al. In contrast to the PTR data, the ozone induced enhancement of the GC signal increases with acetaldehyde concentration. This effect can be due to the different materials used for the tubing inside the instruments: Deming et al. showed, that in glass and metal tubing competitive adsorption occurs, which depends on humidity, concentration and functionality of the analyte, while polymer tubing shows independent absorption. Our fast GC instrument is equipped with heated silico-steel tubing, which allows competitive adsorption, while the PTR is equipped with PFA (perfluoroalkoxy alkanes) tubing. Additionally, with increasing $O_3$ mixing ratios ozonolysis reactions during trapping are gaining importance. It seems, that the interferences on the VOC measurements caused by high ozone exposure are an effect of both, inlet line and instrument's surfaces. The higher GC-signals at 2 ppb of calibration gas are again assigned to the difference in sensitivity (due to filament degradation) already mentioned in the previous section. In Figures 3 and 4a, the acetaldehyde signals rise significantly when 1000 ppb of ozone was present. For the zero air sample (Figure 4b) the signal rose after increasing ozone from 150 ppb where the effect was negligible to 400 ppb. The acetaldehyde signal increased further between 400 and 1000 ppb $O_3$ to about 0.4 ppb (GC) to 0.5 ppb (PTR). Interestingly, the GC signal in Figure 4b did not drop when the standard gas level dropped to 0 ppb. This is an interesting observation that we currently cannot explain. No abnormal behavior in the GC-MS could be ascertained at this time including retention time shifts, tuning anomalies or changes in RH. We conclude that most likely it was an unlogged flow switching issue. Nevertheless, it does not interfere with our general observation, that the acetaldehyde signal is suffering positive interference under high ozone exposure, most likely due to ozonolysis reactions at the tubing surface.

Besides acetaldehyde, C3 and C4 aldehydes and ketones have also been measured, namely propanal, acetone, butanal and MEK. Unfortunately, for those compounds the results were not as clear as for the C2 carbonyl described above. The PTR-ToF-MS in $H_3O^+$ mode cannot separate the aldehyde from the ketone as they have exactly the same mass, i.e. the PTR-Tof-data presented here always shows the sum of propanal and acetone (C3), and butanal and MEK (C4) respectively and should be therefore double the GC signals for the separated species. When measuring zero air, ozone increases the signal of the C3 and C4 carbonyls (cf. Figure 5), starting at $O_3$ mixing ratios of 400 ppb, similar to acetaldehyde. This is most likely due to the reaction of ozone with species attached/adsorbed at the walls of the instrument sampling systems or to unmeasured ozone

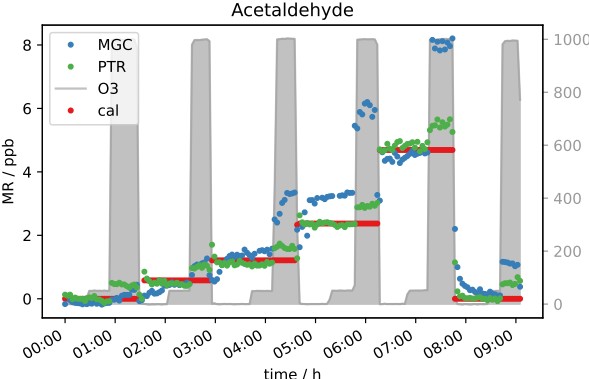

**Figure 3.** Five different standard gas levels between 0 and approx. 4 ppb at ozone mixing ratios of 0, 50 and 1000 ppb.

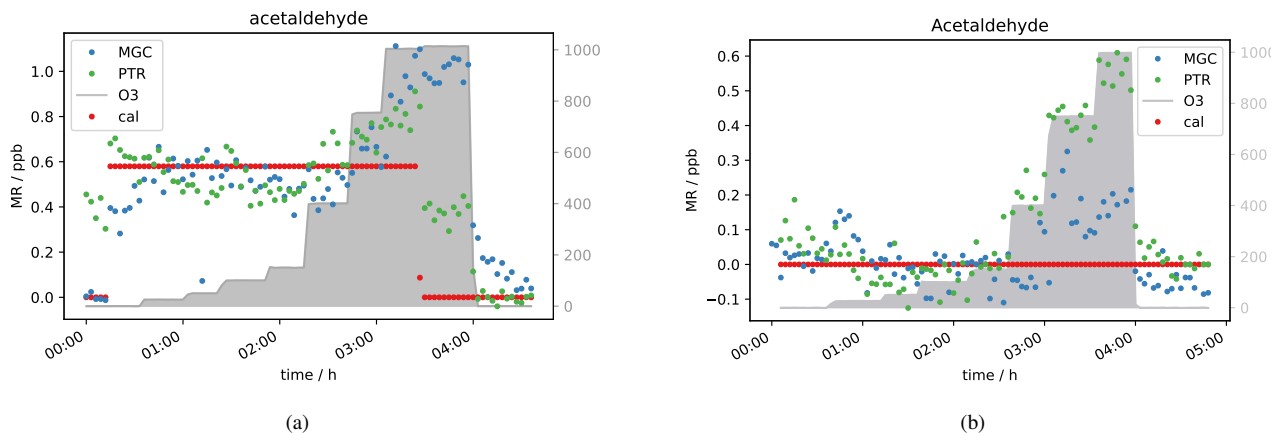

(a)                                                                                  (b)

**Figure 4.** Acetaldehyde mixing ratios at seven different ozone levels between 0 and 1000 ppb and one standard gas level. VOC levels are 0 ppb (a) and 0.5 ppb (b).

reactive species in the zero air. While butanal (measured by GC-MS) shows a strong increase, MEK does not show significant increase in the GC data.

Interestingly, the aldehyde mixing ratios are relatively stable with a tendency to decrease with ozone when the standard gas was added. Figure 6 shows this phenomenon. Propanal and butanal mixing ratios do not show a substantial increase under the same $O_3$ conditions where they increase in the zero air measurement, while the sum of C3 carbonyls (PTR signal) and GC acetone again increase (as in the zero air measurement). As propanal slightly decreases and acetone strongly increases with ozone, the PTR measurements shows a positive net ozone effect for the C3 carbonyls. For C4 carbonyls, the GC quantification during this experiment was compromised (too low mixing ratio) for unknown reasons. However, the qualitative results match the rest of our observations: butanal decreases slightly, while MEK increases slightly, leading to a stable signal for the sum of butanal and MEK, which is shown by the PTR data presented in Figure 6b. Additionally, the qualitative results of butanal and

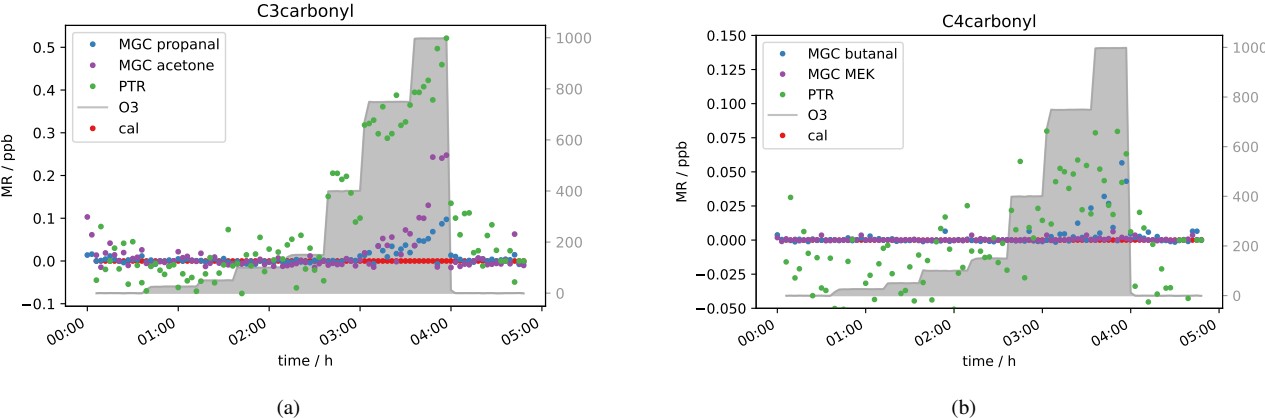

**Figure 5.** C3- and C4 carbonyl mixing ratios of a zero air sample at different ozone levels.

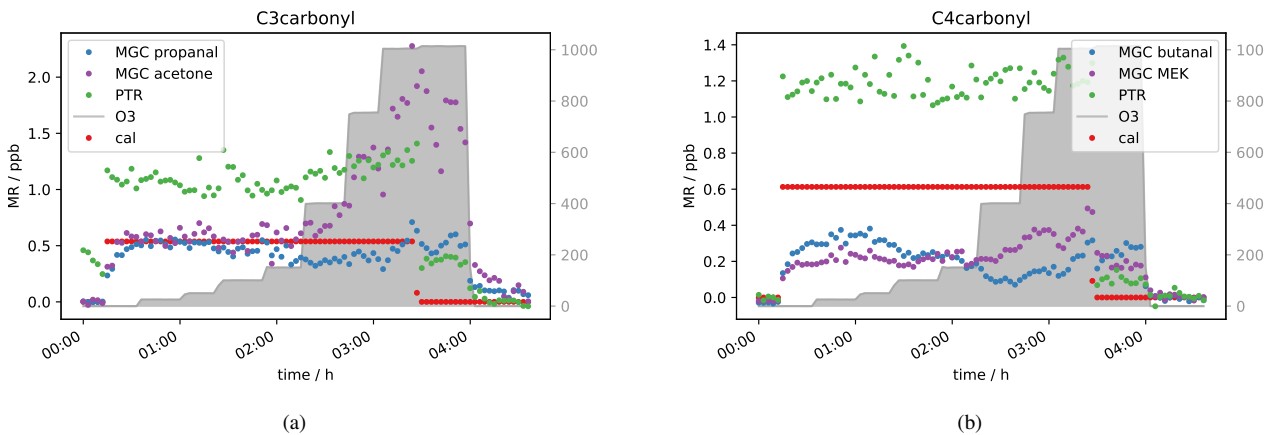

**Figure 6.** C3- and C4 carbonyl mixing ratios at approximately 0.5 ppb per VOC at different ozone levels.

MEK are in line with the qualitative results of C2-C3 carbonyls: The signals for the ketones (acetone (C3), MEK (C4)) increase
with $O_3$ mixing ratios ≥400 ppb and the signals for the aldehydes (acetaldehyde (C2), propanal (C3) and butanal (C4)) are
relatively stable with a tendency to decrease between 200 and 400 ppb $O_3$ and increase as well with $O_3$ mixing ratios ≥400 ppb.
Between 3.5 and 4 h after start of the experiment (cf. Figure 6), not all signals dropped to background levels. They finally drop
once ozone was switched off. This is consistent with the results from the zero air measurement (Figure 5) and the acetaldehyde
data (Figure 4b). It shows that exposure of the inlet tubing to high ozone does not rapidly clean the lines of the interfering
compounds. Apel et al. reported carbonyl generation in the presence of ozone from the rotor material of VICI valves and a KNF
Teflon pump included in their system (Apel et al., 2003). Positive artifacts were reported for acetaldehyde, propanal, acetone
and butanal, they did not find any interference for MEK. This matches our results. We assume that C2-C4 carbonyl compounds
are generated inside the setup inlet tubing and were measured by the GC as otherwise the signal would have dropped when

measuring zero air with 1000 ppb $O_3$. However, at 0.5 ppb of calibration gas we observed a C3 and C4 aldehyde loss under high ozone exposure in the GC measurements. A possible explanation for this is the effect is due to OH radicals produced via Criegee intermediates from the ozonolysis reaction of alkenes present in the standard gas. OH radicals react preferentially with aldehydes rather than ketones. Apparently, these reactions are faster than the C3 and C4 aldehyde generation within the sampling setup. All the experiments show the same trend: under dry conditions, at tropospheric ozone levels ($\leq$ 150 ppb $O_3$), no interference from the oxidant can be observed, while under stratospheric conditions (> 400 ppb $O_3$) there is a strong interference. This leads to the conclusion, that both instruments can measure C2 to C4 carbonyls in the troposphere without being restricted by the amount of ozone present. No reliable results can be achieved when ozone is above 150 ppb under dry conditions. Hence, the current setup is not suitable for stratospheric measurements of these oxygenated species.

### 3.1.3 Ozone causing negative interference

Due to the presence of reactive double bonds, terpenes (isoprene, alpha pinene and beta caryophyllene) were expected to show a decrease in mixing ratio with increasing ozone concentration. This was indeed observed. In Figure 7 it can be seen that the monoterpene signal at 50 ppb of $O_3$ is ten to twenty percent lower for each VOC level compared to the signal without ozone present. The signal drops by almost the same percentage when $O_3$ was increased to 1000 ppb. For sesquiterpenes the effect is even stronger. With 50 ppb $O_3$, the signal drops by roughly 80 %, while the signal is close to or below detection limit at an ozone level of 1000 ppb depending on the terpene mixing ratio applied. This behavior can be explained with reference to the different reaction rate constants $k$ of the mono- and sesquiterpenes in the calibration gas with ozone. $\alpha$-pinene, which is the monoterpene included in the standard gas mixture, has a reaction rate constant with ozone of 9.6 x $10^{-17}$ $cm^3 molecules^{-1}s^{-1}$ while $k$ for $\beta$-caryophyllene is 1.2 x $10^{-14}$ $cm^3 molecules^{-1}s^{-1}$ (IUPAC, 2021). With a reaction rate more than three orders of magnitude larger, the reaction is fast enough to remove 1 ppb of $\beta$-caryophyllene within the short time the sample air needs to travel through the inlet line (< 10 s). Furthermore, it can be seen that the sesquiterpene needs considerable time (more than an hour) to reach a steady state level in the beginning of the experiment ($\sim$ 03:00 in Figure 7), even with lines heated to 45 °C. This could be due to independent absorption of the FEP tubing of these less volatile species (Deming et al., 2019).

Another compound which was expected to show a decreasing mixing ratio with increasing ozone concentration due to its double bonds was isoprene. It was measured simultaneously with PTR- and GC-MS. As can be seen in Figure 8, the signal is not measurably affected by adding 50 ppb of $O_3$. At an ozone level of 1000 ppb, the GC detects roughly 50 % less isoprene. This fits with expectations as isoprene ozonolysis is rather slow, the reaction rate coefficient being close to the one for $\alpha$-pinene (1.3 x $10^{-17}$ $cm^3 molecules^{-1}s^{-1}$), which results in only partial depletion of the isoprene present. Interestingly the PTR-ToF shows slightly higher mixing ratios of the isoprene mass (*m/z* 69) when ozone is present. The elevated signal on this mass can be caused by carbonyl compounds present in sample air or inlet line. Literature reports the same exact mass commonly used for isoprene detection in PTR systems to be a fragment of certain aldehydes (Buhr et al., 2002; Ruzsanyi et al., 2013). Most likely the PTR *m/z* 69 signal in the present study is elevated because under the experimental conditions the positive offset from the carbonyl compounds is higher than the isoprene depletion. For the GC-MS, aldehydes do not interfere as the analytes are separated in the chromatographic system prior to detection. The assumption is supported by the measurement of a zero air

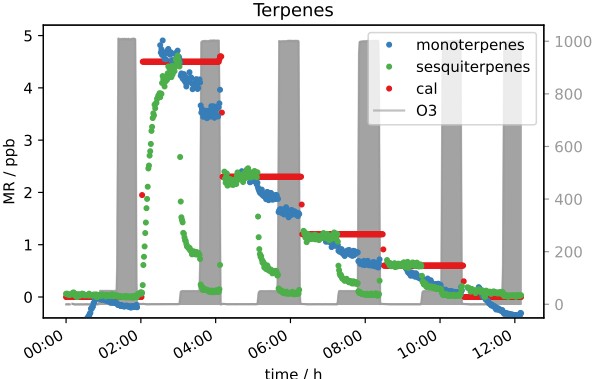

**Figure 7.** Different terpene levels at 0, 50 and 1000 ppb O$_3$ measured with PTR-MS.

sample (no standard gas added into the system) at different ozone levels (see Figure 9a). The GC does not detect any isoprene in the zero air sample regardless of the ozone concentration, while the PTR-MS shows an increasing m/z 69 signal at O$_3$ levels
between 150 and 1000 ppb. Apparently the PTR-ToF-MS signal correlates with the oxidant's concentration. At 150 ppb O$_3$ the PTR-MS detects approximately 100 ppt of signal at the isoprene mass, at 400 ppb of ozone PTR measurements show around 150 ppt and with the highest tested O$_3$ concentration (1000 ppb) roughly 350 ppt.

Figure 9b shows data obtained from the measurement of one isoprene level (∼0.6 ppb) at seven different ozone mixing ratios and without ozone. GC-isoprene is stable until 100 ppb of O$_3$. Somewhere between 150 and 400 ppb the isoprene signal
starts to decrease until it reaches approximately two thirds of the real value at 1000 ppb O$_3$. The two analytical instruments show reverse effects of ozone being present: the PTR-isoprene mass signal is increasing over the same range as the GC signal decreases. This is in line with the measurement of the zero air. The PTR-signal increase is probably due to fragments of other compounds on the same exact mass. In literature, increased PTR *m/z* 69 signals have been reported in high ozone environments in indoor and outdoor air studies which could not be attributed to isoprene (Colomb et al. (2006), Wang et al. (2022)). Those
studies might have been influenced by the same interference we observed.

### 3.1.4   Effect of sodium thiosulfate scrubber on VOC measurements

Generally, when the sodium thiosulfate impregnated quartz filters were added to the sampling system (see Figure 1), no influence on the VOC measurements was seen for the selected VOCs. The literature also reports that this scrubbing material is suitable for measurement of many VOCs without causing interferences (Helmig, 1997; Pollmann et al., 2005). In Figure 10
the measured terpene mixing ratios 0 and 2 ppb VOC with and without scrubber are shown. It can be seen, that the scrubber performed as expected and removed the ozone effectively. α-pinene levels reach the same concentration with the scrubber as before ozone was applied. The measurement of this compound is not affected by the sodium thiosulfate filters and if the scrubber is applied, there is no interference on the measurement with up to 170 ppb of ozone. As mentioned before β-caryophyllene again needed a long time to reach steady state when 2 ppb of the VOC were applied to the system (inlet line temperature

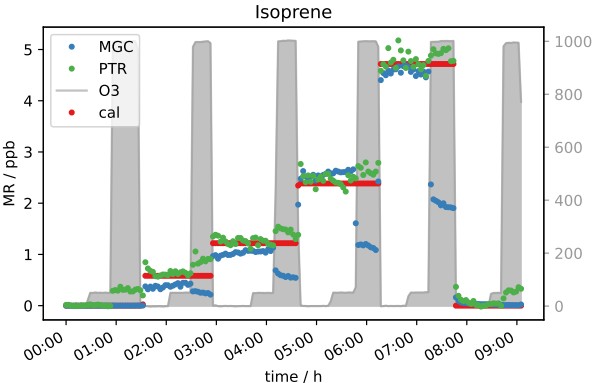

**Figure 8.** Different isoprene levels at 0, 50 and 1000 ppb $O_3$.

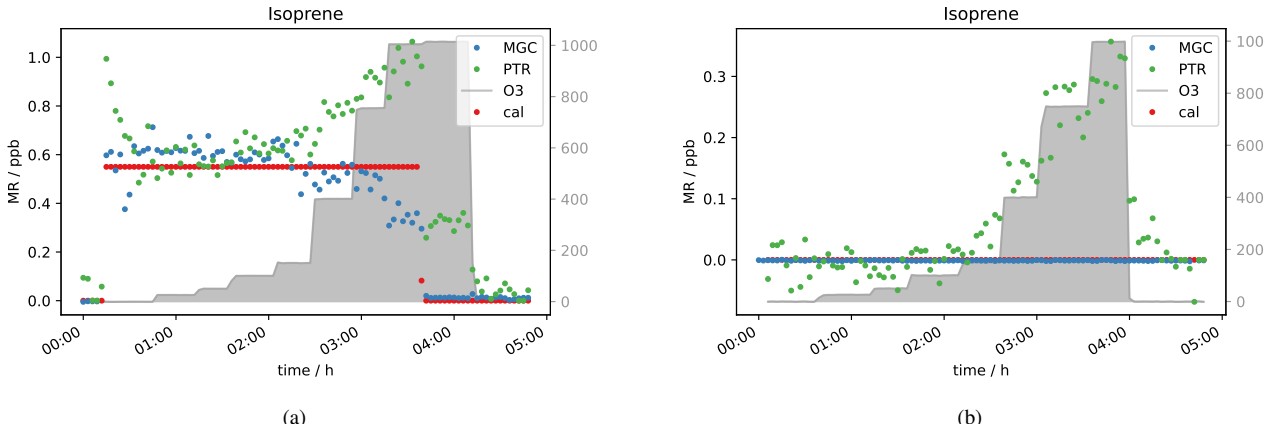

| (a) | (b) |

**Figure 9.** Isoprene mixing ratio at different ozone levels.

45 °C, cf. section 2.1). Steady state is reached after almost two hours when the scrubber was introduced (∼04:30). For the 30-40 min when the thiosulfate filters were introduced (∼4:50-5:30) steady state is not even reached. Sesquiterpene adsorption by the tubing material would again explain this effect. The scrubber was connected using Teflon tubing and a filter holder of the same material. The first time when the terpene-rich air was directed through the scrubber (50 ppb $O_3$, five hours after start of the experiment, (∼4:50-5:30)) the sesquiterpene mixing ratio increased within half an hour, while later (170 ppb $O_3$,

(∼6:00-7:30)) this was not the case. In other words, in the first half hour the sesquiterpenes appeared to be absorbed by the tubing (∼4:50-5:30), while afterwards the material was conditioned and the concentration could reach steady state (∼6:00-7:30). Steady state mixing ratios decrease with increasing $O_3$ mixing ratio as the scrubber was installed roughly half way between the junction where the standard gas was connected and the analytical instruments. On their way to the scrubber, the sesquiterpenes are depleted by ozonolysis. Furthermore, at the applied flow rate of approximately 650 sccm, there is already about 10 ppb of

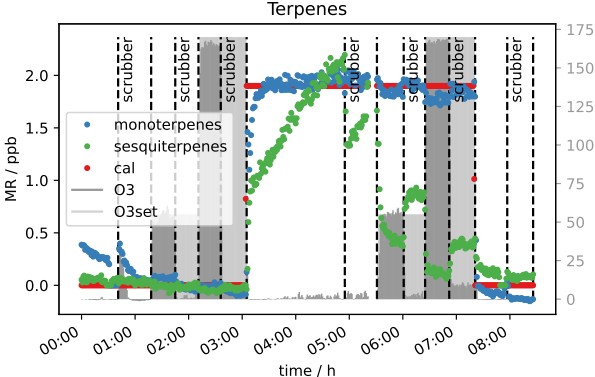

**Figure 10.** Terpene mixing ratios measured by PTR-MS with and without scrubber at 50 and 170 ppb $O_3$.

ozone passing the scrubber (cf. section 3.2) which allows the oxidant to further react with β-caryophyllene on the way between the scrubber and the detector. An improved filter assembly or several scrubbing filters as suggested by Pollmann et al. could improve the sesquiterpene sampling. For all other compounds including isoprene no interference from the scrubber itself could be observed. Furthermore, interferences from ozone on the measurements of analytes like isoprene or acetaldehyde could be eliminated with the filter scrubber in line. The effect on sesquiterpenes could probably also be eliminated if the scrubber was placed at the inlet's front end and the flow through the scrubber was lower. However, it cannot be excluded that the sesquiterpene signal is affected by the scrubber material. Therefore, a longer time span with the scrubber in the sampling line would be required for the signal to reach steady state and the instrument would be unresponsive to rapid changes.

## 3.2 Scrubber endurance

Figure 11 shows the $O_3$ mixing ratio measured with the ozone monitor behind the sodium thiosulfate scrubber under the dry condition at approximately 200 sccm. Results from the experiment at 1000 ppb $O_3$ can be seen in Figure 11a. The ozone concentration behind the filter never reached 0 ppb, already at the beginning 100 ppb of the oxidant could pass the scrubber. That means 1000 ppb of ozone result in an overload of the scrubber capacity, and it was not suitable to remove the oxidant completely from the dry sample air - not even for short time periods. The concentration increases slightly in the beginning, while after 15 h a sudden signal rise can be observed where the mixing ratio increases from roughly 200 ppb to 500 ppb. After 55 h no ozone was scavenged anymore, the concentration before the scrubber is the same as after. Despite inefficient scrubbing under these conditions, the scrubber lifetime would be 15 h as after this time period a strong increase in the signal of the ozone monitor can be observed. In total it can be concluded that the sodium thiosulfate impregnated quartz filters made in the way described in the method section are not suitable for measurements at these flow rates and such high ozone levels as encountered in the lower stratosphere due to insufficient $O_3$ removal and relatively fast depletion of the scrubbing material. However, at lower flow rates it might be more efficient and effective.

Figure 11b presents the result from a similar experiment but with 150 ppb of ozone applied, which corresponds to high ozone

levels at ground level, and typical for the upper troposphere. In the very beginning, the $O_3$ concentration was already 20 ppb. It can be excluded that the filter is not able to remove the $O_3$ completely as the experiment was started twice under these conditions. The first time the ozone mixing ratio was zero at the beginning, but unfortunately the experiment had to be stopped after 50 h. There was no sudden signal increase visible within that time. This 20 ppb offset during the second experiment might be caused by the filter not being perfectly centered in the filter holder. It is possible that a tiny stream of air bypassed the filter inside the filter holder as the 37 mm quartz filter was placed under a 47 mm Teflon filter as described in section 2.3. With 150 ppb of ozone applied, the measured mixing ratio increased by 20 ppb within 40 h and up to 50 ppb within 80 h. As previously shown even isoprene was not affected by 50 ppb of ozone. The only compounds affected by such low $O_3$ levels were mono- and sesquiterpenes. Therefore, it is concluded that the ozone scrubber can be used up to 80 h (3.3 days) at a flow of 255 sccm. The scrubber lifetime test for the lowest ozone concentration used here (50 ppb) took in total more than 20 days. During the first 300 h (12.5 days) the signal didn't change and was below or close to the limit of detection of 0.5 ppb. Around 300 h the measured mixing ratio started to be above detection limit, but still below 5 ppb $O_3$. A sudden increase occurred after approximately 350 h (14.5 days), where the mixing ratio climbed up to 20 ppb within 1.5 days. After the total time of 477 h (almost 20 days) the experiment was stopped, the mixing ratio after the scrubber reached more than 30 ppb. The breakthrough point of the $Na_2S_2O_3$ impregnated filter after 350 h was determined as the end of the scrubber lifetime i.e. at background $O_3$ mixing ratios of 50 ppb the scrubber lasts more than 14 days.

Results of the test at a flow of 550 sccm, which corresponds approximately to the flow used for the experiments to investigate the effect of ozone on the VOC measurements (section 3.1) are presented in Figure 12. Please note that for the average and minimum scrubber lifetimes only measurements at $O_3$ levels 150 and 1000 ppb have been included as for 50 ppb there was no measurement at approximately 550 sccm. For quality assurance it is considered important to have a real measurement which can be compared with the calculated results.

The scrubber lifetime is 8 h at 550 sccm and 1000 ppb $O_3$ according to Figure 12a. At the lower ozone level (150 ppb), the actual flow was 620 sccm. From the plot a lifetime of 30 h has been determined.

Scrubber lifetime $\tau$ highly depends on ozone concentration and flow rate. In Table 3 measured flows and lifetimes can be found. The data was fitted with a power function (y=ax$^b$, cf. Figure 13) being the best fit with $R^2$ close to 1. Additionally, the lifetimes for exactly 200 and 550 sccm were calculated with equation 1 and can be found in Table 4. The factors *a* and *b* obtained from the fit were rounded to 3500000 and -1 respectively as small changes are attributed to measurement uncertainties.

$$\tau = \frac{3500000}{O_3 * flow} \tag{1}$$

The ozone levels were chosen to correspond to those experienced in the troposphere (50 and 150 ppb $O_3$) and stratosphere (1000 ppb $O_3$) so that replacement times of the scrubber under field conditions could be determined. Flows correspond to the inlet flows of the mass spectrometers (200 sccm) and to the flow used for the experiments described in section 3.1.

The calculated lifetime at 200 sccm and 1000 ppb of ozone is 18 h. Hence, this is the time after which the filter should be exchanged under these conditions if reducing ozone to 100-200 ppb is enough for the compounds of interest. If the back-

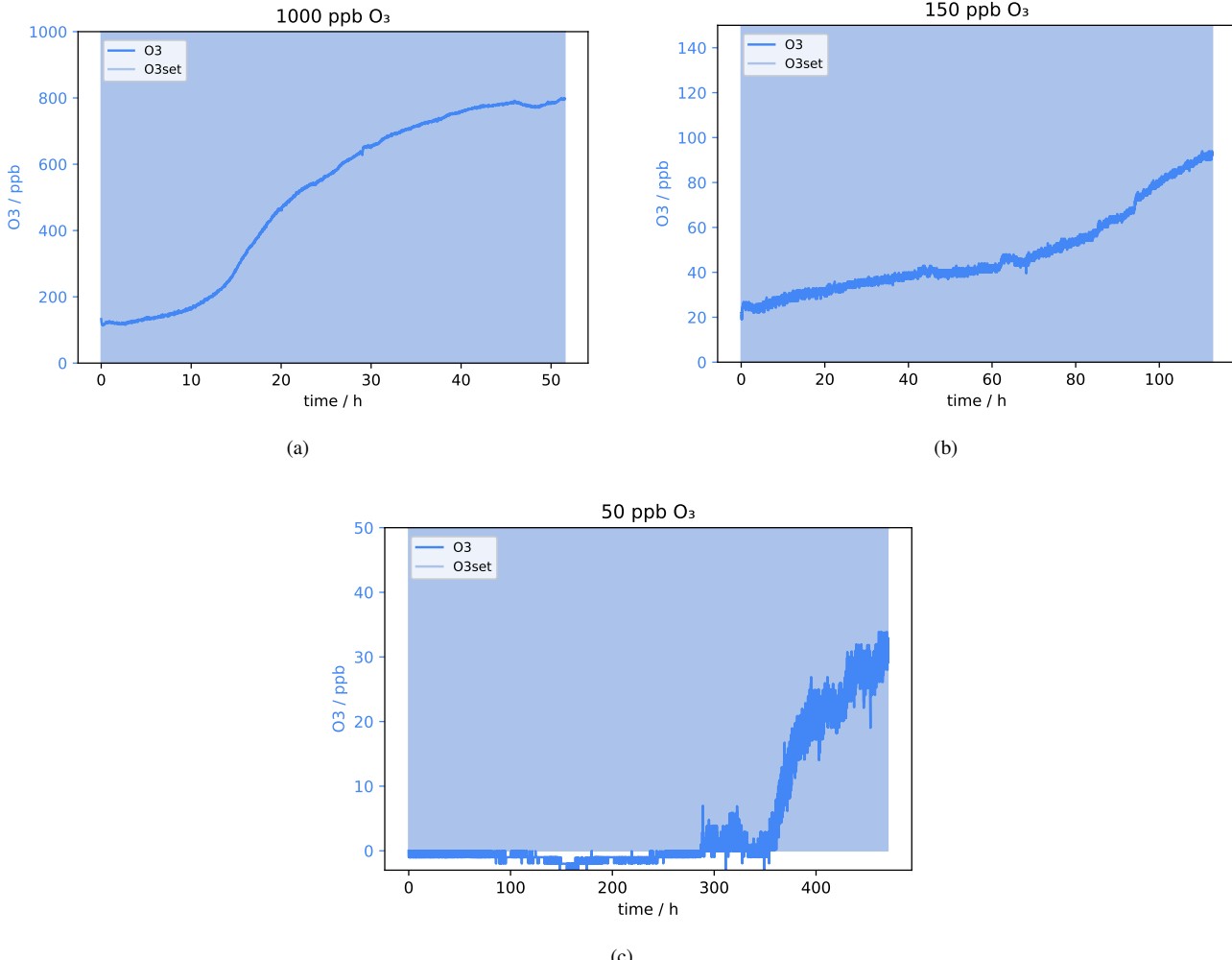

**Figure 11.** O$_3$ mixing ratio after Na$_2$S$_2$O$_3$ scrubber at 0 % RH. Flows and O$_3$ levels before scrubber: (a) 220 sccm, 1000 ppb; (b) 255 sccm, 150 ppb; (c) 230 sccm, 50 ppb.

ground ozone mixing ratio does not exceed 150 ppb and the flow is the same as used for the here applied mass spectrometers (200 sccm), the scrubber needs to be replaced after 117 h to avoid interference from ozone for analysis of most VOCs except terpenes. However, on many ground based measurement sites in the troposphere the ozone mixing ratio rarely exceeds 50 ppb, which would result in a scrubber exchange every 14 days to assure efficient ozone removal. The lifetime calculation at 550 sccm and stratospheric ozone concentration results a scrubber lifetime of 6 days and is slightly lower than the measured lifetime (8 days, cf. Figure 12a). With 150 ppb O$_3$ and the same flow $\tau$ would be 42 h and filter replacement due in less than 2 days. The determination of the lifetime is approximate as we use the time at which a sudden increase in ozone concentration after the scrubber was observed. Exchanging the filter earlier does not affect the data quality and is therefore recommended.

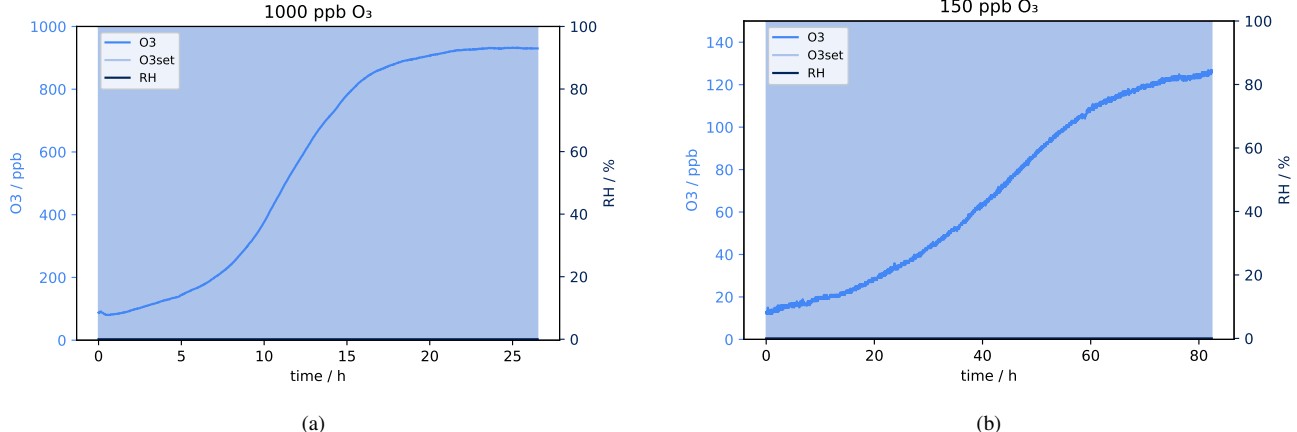

**Figure 12.** $O_3$ mixing ratio after $Na_2S_2O_3$ scrubber at 0 % RH. Flows and $O_3$ levels before scrubber: (a) 550 sccm, 1000 ppb; (b) 620 sccm, 150 ppb.

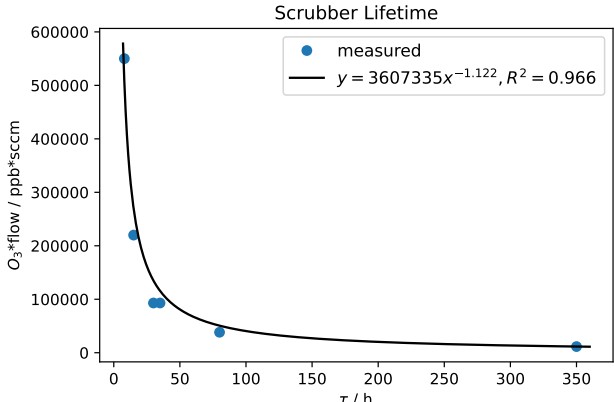

**Figure 13.** Measured data from the scrubber endurance test.

The filter assembly could be improved by using a filter housing which fits perfectly to the filter diameter in order to avoid any small air stream bypassing the filter. In this study, we did not examine the effect of several filters placed in series in the line as it was done by Pollmann et al. They found that with additional filters the scrubbing efficiency and scrubber endurance could be improved, but we adopted single filters to test their efficacy and endurance while minimizing potential uptake losses.

However, in section 3.1 the presented experiments were all performed in less than one day and the maximum ozone concentration (1000 ppb) was applied for only several hours. Thus, the ozone concentration was stable throughout each experiment.

**Table 3.** Ozone mixing ratio and corresponding flows through the scrubber during scrubber endurance tests as well as resulting lifetimes τ.

| O$_3$ / ppb | flow / sccm | measured | |
| | | τ / h | τ / days |
|---|---|---|---|
| 1000 | 220 | 15 | 0.6 |
| 150 | 255 | 80 | 3.3 |
| 50 | 230 | 350 | 14.6 |
| 1000 | 550 | 8 | 0.3 |
| 150 | 620 | 35 | 1.5 |

**Table 4.** Calculated scrubber lifetimes τ at exactly 200 and 550 sccm.

| O$_3$ / ppb | flow / sccm | calculated | |
| | | τ / h | τ / days |
|---|---|---|---|
| 1000 | 200 | 18 | 0.7 |
| 150 | 200 | 117 | 4.9 |
| 50 | 200 | 350 | 14.6 |
| 1000 | 550 | 6 | 0.3 |
| 150 | 550 | 42 | 1.8 |

## 3.3  Effect of humidity

Field measurements can take place in various locations and environmental conditions. Relative humidity may influence the analysis of water soluble compounds. However, the presence of water may also affect the behaviour of VOC molecules when interacting with surfaces. The scrubber material used here is inorganic and water soluble and has therefore been tested under dry and humid conditions. For the measurements performed within this study, humidity did not have any influence on the GC- and PTR-ToF-MS instrument measurement capability as these dried the air before detection (GC-MS) or used humid calibrations (PTR-ToF-MS). Thus any effects observed can be ascribed to the inlet system. Interestingly, the scrubber lifetime increased dramatically at 80 % relative humidity. The test was done twice to double check the results obtained. Figure 14 shows the ozone mixing ratio and relative humidity measured at 1000 ppb O$_3$ and a flow of 230 sccm. RH was relatively stable throughout the whole experiment. As mentioned in section 3.2, under similar dry conditions the sodium thiosulfate scrubber was able to remove only 90 % of the ozone and filter performance dropped drastically after 15 h. In contrast, at 80 % RH the ozone could be removed completely from the sample air. The O$_3$ mixing ratio did not change over 65 h of experiment, it was always below 10 ppb. After 65 h the experiment was stopped.

The only signal that was still influenced by ozone when the filter scrubber was in the sample line, was the sesquiterpene signal. Consequently the corresponding mixing ratio can be used to investigate whether or not relative humidity changes

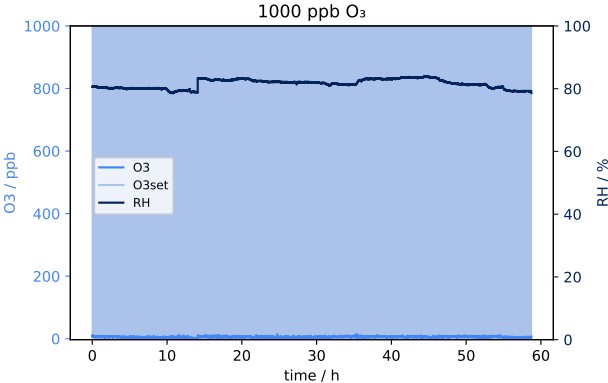

**Figure 14.** $O_3$ mixing ratio after $Na_2S_2O_3$ scrubber at 80 % RH, 230 sccm.

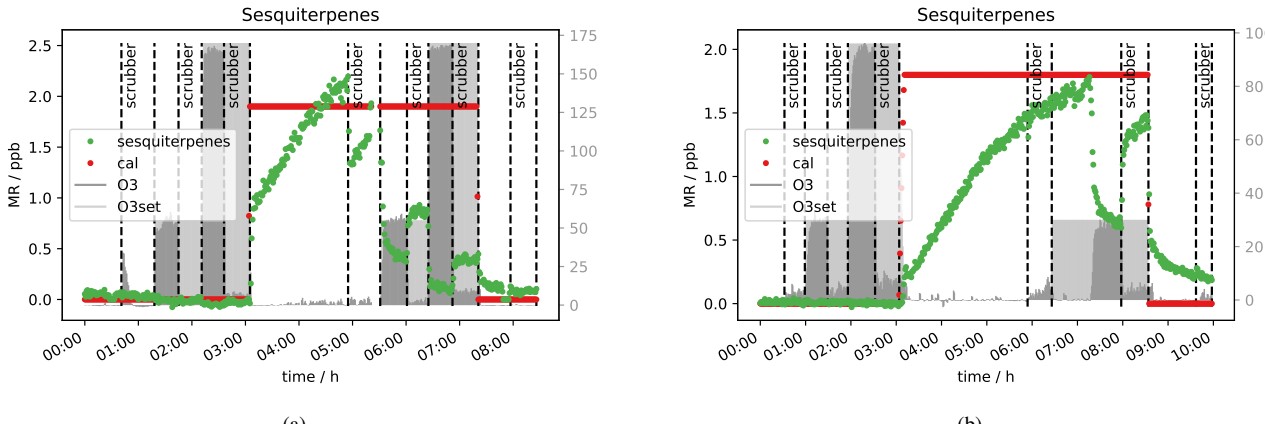

(a)  (b)

**Figure 15.** Sesquiterpene mixing ratio with and without scrubber different $O_3$ mixing ratios when 2 ppb of sesquiterpenes were introduced to the system; (a) 0 % RH, (b) 50 % RH.

the results compared to the dry condition. Figure 15 shows the sesquiterpene mixing ratio under dry (15a) and humid (15b) conditions when ca. 2 ppb standard gas was introduced to the system. In the first plot the signal with scrubber at 50 ppb of ozone (approx. 06:00-06:30) comes back to 50 % of the original signal, while in the second plot it comes back to approx. 80 % (approx. 08:00-09:00) of the original height. This is consistent with the results from the scrubber lifetime tests at 80 % RH and suggest that scrubber performance benefits from increased humidity.

Sodium thiosulfate reacts with ozone to tetrathionate, water and oxygen:

$$2\ S_2O_3^{2-} + O_3 + 2\ H^+ \rightleftharpoons S_4O_6^{2-} + O_2 + H_2O \tag{R1}$$

As it is an equilibrium reaction, the equilibrium will shift to the left if there is an excess of water, recovering the thiosulfate from the $S_4O_6^{2-}$ produced. This could explain why the ozone level during the scrubber performance test at a relative humidity

of 80 % was relatively stable throughout the whole measurement time which was almost 3 days and why all the oxidant can be scavenged at an $O_3$ level of 1000 ppb.

**4 Conclusions**

Most species tested were not affected by ozone being present. Nevertheless, higher mixing ratios were observed for carbonyl compounds indicating an ozone interference. Signals for acetaldehyde, propanal, acetone and butanal increased with ozone levels above 150 ppb in zero air measurements with GC as well as PTR. Thus, it can be concluded, that there are positive artifacts generated in the experimental setup i.e. in the tubing, inside the ozone generator or within both of the mass spec-

415 trometers. As we observed the same during stratospheric measurements before, the ozone generator cannot be the only source. Apel et al. originally conducted ozone sensitivity tests on their airborne GC-MS system due to anomalous acetaldehyde observations in the stratosphere. Our experiments also show acetaldehyde to be the species most affected by ozone interference. When our PTR encountered a stratospheric intrusion in flight as on $2^{nd}$ June 2020 we found 0.88 ppb acetaldehyde (altitude 13000 m, $O_3$ 465 ppb), extremely suspect for such a short-lived molecule under otherwise clean conditions. Unfortunately, the

420 fast GC-MS did not measure acetaldehyde during this flight campaign. When 0.5 ppb of VOC standard gas was measured, the signals for propanal and butanal decreased due to the reaction with the OH radical, generated from ozonolysis of terpenes. For MEK no ozone interference could be observed. Decreasing signals were found for monoterpenes ($\alpha$-pinene), sesquiterpenes ($\beta$-caryophyllene) and GC-isoprene as expected due to the direct reaction with ozone. The PTR-ToF-MS measured increasing "isoprene" mixing ratios on *m/z* 69 with increasing ozone concentration due to aldehyde fragments on the same exact mass.

Those fragments are most likely ozonolysis products from some adsorbed species inside the FEP tubing or the instrument itself from some former measurements. Therefore caution should be applied when interpreting the isoprene mixing ratios provided by PTR-ToF-MS in high ozone environments. Generally the scrubber lifetime under tropospheric conditions ($O_3$ mixing ratios between 50 and 150 ppb) and a flow of 200 sccm is between five and fourteen days. The inlet flow of both instruments is the same (200 sccm), therefore the result can directly be used for ground based field measurements with the here employed

PTR-ToF-MS and fast GC-MS systems. Sodium thiosulfate impregnated quartz filters are a very convenient means of ozone scavenging as the filters can be prepared in advance easily, at low cost and can be stored in a dry, dark and clean place. The preparation as well as the application are simple and the usage time for the filters is with several days long enough to allow measurement of complete diel cycles without stopping the measurements in between to open the inlet line for scrubber exchange. Nevertheless, the scrubbing technique as applied here is not suitable for sesquiterpene measurement. The observed

loss of these analytes could be reduced with a $Na_2S_2O_3$ scrubber in line, but it could not be eliminated completely. Similar results have been reported earlier. Pollmann et al. (2005) found ozone concentrations in the sampling air reduced to 0.4 % when using a thiosulfate scrubber at an ozone mixing ratio of 100 ppb and 255 sccm. Unfortunately they did not report how long these conditions were measured. Nevertheless, their finding is comparable to that reported here, at 150 ppb $O_3$ and at the same flow the scrubber could reduce ozone to almost zero while at 50 ppb the oxidant's mixing ratio was below the limit of detection

for more than 14 days. The tests have shown that the scrubbing technique used in this study performs well for most ground

based operations of the VOC instruments, as inlet flows and ozone levels are usually low enough to stay within range of good performance of the sodium thiosulfate filter scrubber. Nonetheless, it is not possible to completely avoid sesquiterpene loss due to reaction with ozone during sampling. The current setup is not suitable for measuring highly ozone reactive compounds such as sesquiterpenes or under high ozone conditions like in the lower stratosphere. Fortunately, since sesquiterpenes are emitted from the surface, they are extremely unlikely to be present in the stratosphere. In general it can be concluded that higher humidity has a positive influence on the VOC measurements performed here with a sodium thiosulfate filter scrubber, when ozone was present and no interference from the humidity on the gas analysis itself could be observed. Still, the $O_3$ removal is not sufficient to measure sesquiterpenes when ozone is present, not even at low mixing ratio such as 50 ppb. Implementing a stable humidification to the measurement system is non-trivial and additional effects for other analytes could develop. It is important to continue the development and characterization of reliable techniques which are cheap and easy to implement for lab as well as field experiments.

In summary we can say that insertion of the ozone scrubber resulted in the removal of most of the interferences observed. This implies that most of the effects observed were initiated in the inlet and any residual effects were produced within the instruments being therefore different and specific for each instrument. It is important to note that these improvements apply to the suite of gases tested here, and presumably also to those with comparable vapor pressures and ozone reactivities. The filter system could be further improved with a low dead volume filter housing to avoid any tiny air streams bypassing the filter inside the assembly and by the installation of multiple scrubbing filters in series as was tested by Pollmann et al. However, it is important to consider that the introduction of a filter into the system can also induce some negative effects. For example, highly oxygenated low volatility species are likely to suffer losses on such a filter assembly. Such compounds may need entirely different approaches such as inlet-less collection onto adsorbent filled cartridges or ozone removal at the inlet entrance by the addition of nitric oxide (NO). Furthermore, the filter itself can introduce flow rate limits to the inlet due to its physical restriction of flow. Generally, for field studies, our current recommended strategy is to use a high-volume, constant temperature, flow from inlet tip to close to the instrument and then subsample that flow, through the ozone scrubber, into the instrument at a lower rate. The inlet material should be Teflon in agreement with the findings of Deming et al. VOC emitting materials such as silicone should be avoided and during high local pollution events (such as in an aircraft taxiing on the ground) inlets should be stoppered or back flushed to avoid strong contamination.

*Data availability.* The data are published at https://doi.org/10.5281/zenodo.7576413 (Ernle and Ringsdorf, 2022).

*Author contributions.* LE and MAR designed the experiment; LE and MAR performed the measurements; LE and MAR analyzed the data; LE wrote the manuscript draft; MAR and JW reviewed and edited the manuscript.

*Competing interests.* The contact author has declared that neither they nor their co-authors have any competing interests.

*Acknowledgements.* We would like to acknowledge our lab technicians Thomas Klüpfel and Rolf Hofmann for exchanging gas bottles during long experiments.

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
