# Peer review of "Influence of ozone and humidity on PTR-MS and GC-MS VOC measurements with and without $Na_2S_2O_3$ ozone scrubber"

_Atmospheric Measurement Techniques, 2022_

## Referee Comment (RC1)

Review of "Influence of ozone and humidity on PTR-MS and GC-MS VOC measurements with and without Na2S2O3 ozone scrubber by Ernle et al., https://doi.org/10.5194/amt-2022-279

The manuscript tackles an imported artefact associated with in-situ VOCs measurements, especially in polluted (ozone-rich) environments on ground-based sites and the upper troposphere and lower stratosphere. The paper is well structured and focused and a pleasure to read. Congratulations!

Still I have a couple of issues that need to be clarified (see minor concerns) and one general comment.

**General comment:**
By integrating the ozone scrubber just before the two instruments, the observed artefacts are basically gone. This implies that the ozone-driven chemistry occurs within the instruments, namely in their instrument inlet sampling system and the detection system. Both sampling and detection systems will differ quite substantially and to my understanding, the observed effects/artefacts often differ in their magnitude. It would be very helpful for all research groups using the two measurement techniques in the field, if you add a section "lessons learned" (or so) that summarizes your understanding of the problem and that lists your recommendations. I guess you will have learned a lot with the two different instruments and that you can give more advice then "just": install a sodium thiosulfate ozone scrubber. For instance:

1) Do you only expect surface effects (incl. memory effects) on the walls of the sampling system or may also gas-phase reactions (e.g. in the drift tube) play a role? Can these effects be minimized by using special sampling lines, e.g. made of PEEK or (silanated) silcosteel that show much smaller permeation and thus memory effects than lines made of PFA or FEP?

2) Another issue in this respect: All relevant reactions and their magnitude will depend on the cleanness of the instrument. Based on your experiences, can you give relevant advice, e.g. to clean the instrument before starting measurements with 500 ppb ozone for half a day? By how much the ozone-driven artefacts will decrease. Or in other words, with an uncleaned instrument, one can't get reliable data for some target gases such as acetaldehyde or acetone even at lower ozone m.r.? I also ask here because you haven't specified the pre-treatment of your instruments (you should add this, yet). And on L. 178/179 you write that just adding ozone increases the signal of the C3 and C4 carbonyls, most likely (but not written there) because of reaction of ozone with species attached/adsorbed at the walls of the instrument sampling systems.

3) Is the installation of such a scrubber accompanied with any disadvantages, e.g. the affection of certain species or an increased response time (due to memory effects)? If you don't have relevant experiences, you could speculate a bit, e.g. that (based on your understanding) such effects are unlikely or possible for certain species.

4) The effects occur between the location where ozone is added or present and somewhere in the detection chamber/system. During atmospheric measurements the reaction times are usually longer, as ozone enters the sampling line together with the

sample air and then travel in common until the detection system. Please estimate the total reaction (travel) time in your laboratory system so that other instrument users can judge the problem in their configuration.

**Minor concerns**

- General remark: I suggest to use the term "zero air" instead of "background air". The later usually characterizes "not polluted" sample air. Moreover, is your synthetic air really clean or VOC-free, so that its influence on your experiments and results can be excluded?
- L. 73. Please add that ozone (as non-polar) molecule is little affected / solved in the water bubbler
- L. 78. Please use SI units, that is "hPa" instead of "mbar".
- L. 80. How H2O and RH was measured and where? What is your reference temperature for calculating RH, just the laboratory temperature and you assume that the temperature of the scrubber assembly is identical?
- L. 111f (sections 2.2.2 and 2.2.3). Shortly describe the material (FEP, PFA, PEEK, silcosteel,…) and parts (sampling tubes, …) that are in contact with the sample air and on which surface reaction can occur. And what are the residence (reaction) times in your system? See also general comments.
- L. 112. "hPa"
- L. 117. "was" before 2.85 m
- L. 131f. A major topic of your paper is the influence of the ozone scrubber. To better understand its functioning, you should add more information in this section. Do you use a filter holder with an inner diameter of 37 mm (or less)? What is the air residence time in it? Is the scrubber just a "tissue filter" (prepared as you described) or more?
- L. 164f, Fig. 3. Why the enhancements for the GC and the PTRMS are so different?
- L. 168/169. You write "This indicates that the interference is not instrument specific but more likely function of the common inlet tubing exposure to ozone.". However, independent on the filament problem of the GC (that is at 2 ppb), the enhancement between 7:00 and 8:00 (and 1000 ppb ozone) is a factor of ~3.5 higher for the GC. I would conclude that's not only an inlet tube effect or do you expect higher influences by the inlet sampling inside the GC. Please better explain this difference.
- Fig. 4a. What is the reason of the missing signal drop of the MGC at ~3:30 (and the moderate drop of the PTR), when the cal (acetaldehyde) signal jumped to 0? As both signals directly turn to 0 when ozone is switched off, this appears like a reaction of ozone at the surfaces of the sampling lines. Would this also explain the last point, namely the different behavior of both instruments?
- L. 176: "have the same exact mass" → "have exactly the same mass"
- L. 178/179. a) what is the process for the signal increase if ozone is added to zero-air and b) does this process/effect explain the two issues further up?
- Fig. 6. For what species "cal" stands for? … 0.5 ppb propanal and 0.5 ppb acetone, that is, in sum 1 ppb? Then both instruments would measure too little (Fig. 6a and the MGC for C4) or too high (PTRMS for C4). Please better explain this.
- L. 183. "Propanal and butanal mixing ratios decrease under the same O3 conditions …" not so clear in my opinion, because the m.r. appears to increase later in the time series. I would describe this with "indifferent with a tendency to depleted m.r. or so".

- L. 190. "It shows that exposure of the inlet tubing to high ozone does not rapidly clean the lines of the artifact inducing compounds." That's an important sentence and I guess the first time where you describe what's going on. I suggest that you add a further short section 2.5 having a title like "Potential effects causing …" and a short description of a) surface reaction in the sampling lines, b) gas phase reactions in the sapling lines, c) gas phase reaction in the PTR detection system (e.g. in the drift tube or ??) and give them a real process name. Later in following sections you can refer to these three (or more?) "artefact reactions" and (if possible) detail the process further.
- L. 215-217. This part I haven't understood fully. What you mean with "in the very beginning"? After using a new FEP line, after switching to a new air matrix, after …? And what you mean with "independent absorption"? Independent on what?
- L. 217-222. In my opinion, the text starting with "Note that …" fits better in the experimental section.
- L. 227f (and Fig. 8). You say that the carbonyl compounds offset the expected signal drop with ozone or do you believe that the PTR doesn't show a depletion of the isoprene signal (in contrast to the GC) and you "only" see the positive offset from the carbonyl compounds? Please specify and clarify.
- L. 248f (discussion of Fig. 10). Sorry, I can't follow your explanations. First, I couldn't figure out in Fig. 10 when or during which periods the scrubber is connected and when bypassed. Please add bars (or so) at the top. Furthermore, between ~5:00 ad ~7:20 at 2 ppb the sesquiterpenes never reaches the target concentration of 2 ppb, neither with nor without scrubber. I understand that this is due to the absorption/desorption (memory) effects. Still, it's hard to catch your main messages. It's a) that the scrubber has no influence (besides scrubbing ozone) and b) that the sesquiterpenes are strongly affected by memory effects? Is the memory effect limited to the sesquiterpenes and all other species just work fine and are a not affected by the ozone scrubber?
  Please improve your explanations. For a better understanding, it would help a lot adding times or time periods.
- L. 270. "Scrubber endurance" would be a more suited title.
- L. 287-290. These details on the filter assembly belong to section 2.3. I understand that you filter assembly has not been perfect, as you inserted the scrubber tissue into the existing somewhat larger Teflon filter, correct? Can you add here in this section or maybe better in the "lessons learned section" (see general comments) if – based on your experiences – the scrubber/filter assembly can be improved, e.g. by using a suitable filter housing (avoiding bypassing) and more important by adding more scrubber tissues? This will increase the scrubbing efficiency and the scrubber endurance (correct and do you expect a linear scaling with the number of scrubber tissues?), but do you also expect negative effects?
- L. 346f. Only now/here (as you give a time period) I understand the cycle in Fig. 10. In Fig 15b, at 1:00 you switch to O3=50ppb and let this level until 2:00, but at 1:30 you add the scrubber and the measured O3 signal switch to 0, although the O3 level (by the O3 generator) still is at 50 ppb, correct? As requested before, please indicate this cycling in the relevant Figs better.

---

## Author Comment (AC1)

**Reviewer 1**

Review of "Influence of ozone and humidity on PTR-MS and GC-MS VOC measurements with and without Na2S2O3 ozone scrubber by Ernle et al., https://doi.org/10.5194/amt-2022- 279 The manuscript tackles an imported artefact associated with in-situ VOCs measurements, especially in polluted (ozone-rich) environments on ground-based sites and the upper troposphere and lower stratosphere. The paper is well structured and focused and a pleasure to read. Congratulations! Still I have a couple of issues that need to be clarified (see minor concerns) and one general comment.

We thank the reviewer for these positive comments.

General comment: By integrating the ozone scrubber just before the two instruments, the observed artefacts are basically gone. This implies that the ozone-driven chemistry occurs within the instruments, namely in their instrument inlet sampling system and the detection system. Both sampling and detection systems will differ quite substantially and to my understanding, the observed effects/artefacts often differ in their magnitude. It would be very helpful for all research groups using the two measurement techniques in the field, if you add a section "lessons learned" (or so) that summarizes your understanding of the problem and that lists your recommendations. I guess you will have learned a lot with the two different instruments and that you can give more advice then "just": install a sodium thiosulfate ozone scrubber. For instance:

Following the advice of the reviewer we now add the following paragraph to the final section.

l. 446: "In summary we can say that insertion of the ozone scrubber resulted in the removal of most of the artifacts observed. This implies that most of the effects observed were initiated in the inlet and any residual effects were produced within the instruments being therefore different and specific for each instrument. It is important to note that these improvements apply to the suite of gases tested here, and presumably also to those with comparable vapor pressures and ozone reactivities. The filter system could be further improved with a low dead volume filter housing to avoid bypassing of tiny air streams inside the filter assembly and by the installation of multiple scrubbing filters in a row as was tested by Pollmann et al. (2005). However, it is important to consider that the introduction of a filter into the system can also induce some negative effects. For example, highly oxygenated low volatility species are likely to suffer high losses on such a filter assembly. Such compounds may need entirely different approaches such as inlet-less collection onto adsorbent filled cartridges or ozone removal at the inlet entrance by the addition of nitric oxide (NO). Furthermore, the filter itself can introduce flow rate limits to the inlet due to its physical restriction of flow. Generally, for field studies, our current recommended

strategy is to use a high-volume, constant temperature, flow from inlet tip to close to the instrument and then subsample that flow, through the ozone scrubber, into the instrument at a lower rate. The inlet material should be Teflon in agreement with the findings of Deming et al. (2019). VOC emitting materials such as silicone should be avoided and during high local pollution events (such as in an aircraft taxiing on the ground) inlets should be stoppered or back flushed to avoid strong contamination."

1) Do you only expect surface effects (incl. memory effects) on the walls of the sampling system or may also gas-phase reactions (e.g. in the drift tube) play a role? Can these effects be minimized by using special sampling lines, e.g. made of PEEK or (silanated) silcosteel that show much smaller permeation and thus memory effects than lines made of PFA or FEP?

Given the low pressure (2 mbar) and the short residence time of the gases in the drift tube (~0.1 s, cf. Piel, 2021; Innomata, 2008) we do not expect gas phase reactions there to be significant compared to the ambient pressure inlet. In this work we did not vary the inlet materials. The majority of the tubing used in this study was FEP Teflon which was found by Deming et al. 2019 to perform best in a comparison of inlet materials. Although not part of this study we accept the point that the inlet material is an important consideration and we make reference to the work of Deming et al. 2019 as follows:

l. 46: "The majority of the tubing used in this study was FEP Teflon which was found by Deming et al. 2019 to perform well in a comparison of inlet materials (including peek and stainless steel), as adsorption on FEP was found to be independent of humidity, concentration and functionality. The tubing was not new, but used previously for airborne measurements aboard a research aircraft. It was flushed with synthetic air for at least one hour prior to the experiments performed."

2) Another issue in this respect: All relevant reactions and their magnitude will depend on the cleanness of the instrument. Based on your experiences, can you give relevant advice, e.g. to clean the instrument before starting measurements with 500 ppb ozone for half a day? By how much the ozone-driven artefacts will decrease. Or in other words, with an uncleaned instrument, one can't get reliable data for some target gases such as acetaldehyde or acetone even at lower ozone m.r.? I also ask here because you haven't specified the pre-treatment of your instruments (you should add this, yet). And on L. 178/179 you write that just adding ozone increases the signal of the C3 and C4 carbonyls, most likely (but not written there) because of reaction of ozone with species attached/adsorbed at the walls of the instrument sampling systems.

The reviewer raises an interesting point that was not examined in this work, namely the potential effectiveness of cleaning procedures such as introducing high flows of high concentrations of ozone prior to measurement to potentially passivate surfaces.

Northway et al. tested inlet exposure to ozone and found that a kind of passivation occurs after a while. Unfortunately, this was reversible and disappeared during further measurements. As this would lead to a shifting background in the subsequent measurements we do not do this, preferring to rely of the fast equilibration of ambient species to the walls of the fast flow inlet.

For the experiments discussed here, we flushed the tubing for at least one hour with synthetic air (in order to generate similar starting conditions), but without ozone prior to the experiments (cf. line 67).

We now capture these points in the manuscript with the following added text:

l. 68: "When considering the ozone in the instrument inlet, one could consider passivating the inlet surfaces prior to measurement by the introduction of high (500ppb) ozone mixing ratios. Northway et al. tested this possibility and noted a passivation that disappeared during further field measurements. As this will in effect generate a shifting background to the subsequent measurements, and as 6 hour flushing is impractical prior to flight measurements we chose not to follow this procedure. "

Also added at line 178/179:

l. 226: "When measuring zero air, ozone increases the signal of the C3 and C4 carbonyls, starting at $O_3$ mixing ratios of 400 ppb, similar to acetaldehyde (cf. Figure 5). This is most likely due to the reaction of ozone with species attached/adsorbed at the walls of the instrument sampling systems."

3) Is the installation of such a scrubber accompanied with any disadvantages, e.g. the affection of certain species or an increased response time (due to memory effects)? If you don't have relevant experiences, you could speculate a bit, e.g. that (based on your understanding) such effects are unlikely or possible for certain species.

Thank you for this important question. It is theoretically possible that the filter scrubber causes an increased response time due to memory effects and an increased surface area. In our study, which used a single filter, all of the highly volatile compounds show rapid responses when introduced or removed from the gas streams. However, the sesquiterpenes which have the highest mass and ozone reactivity of the compounds tested, do show a slow response with and without filter scrubber (cf. Figure 10). From our measurements, we conclude that the slow response is most likely due to absorption in the Teflon tubing and filter housing as an increasing signal is only observed the very first time, when terpene rich air is directed through a certain tubing (l.309ff). However, it cannot be excluded that this effect is due to both, memory effects of the tubing and the filter material.

These points are included in the response to point one, within the additional lessons learned paragraph.

Text added (as before):

l. 446: "In summary we can say that insertion of the ozone scrubber resulted in the removal of most of the artifacts observed. This implies that most of the effects observed were initiated in the inlet and any residual effects were produced within the instruments being therefore different and specific for each instrument. It is important to note that these improvements apply to the suite of gases tested here, and presumably also to those with comparable vapor pressures and ozone reactivities. The filter system could be further improved with a low dead volume filter housing to avoid bypassing of tiny air streams inside the filter assembly and by the installation of multiple scrubbing filters in a row as was tested by Pollmann et al. (2005). However, it is important to consider that the introduction of a filter into the system can also induce some negative effects. For example, highly oxygenated low volatility species are likely to suffer high losses on such a filter assembly. Such compounds may need entirely different approaches such as inlet-less collection onto adsorbent filled cartridges or ozone removal at the inlet entrance by the addition of nitric oxide (NO). Furthermore, the filter itself can introduce flow rate limits to the inlet due to its physical restriction of flow. Generally, for field studies, our current recommended strategy is to use a high-volume, constant temperature, flow from inlet tip to close to the instrument and then subsample that flow, through the ozone scrubber, into the instrument at a lower rate. The inlet material should be Teflon in agreement with the findings of Deming et al. (2019). VOC emitting materials such as silicone should be avoided and during high local pollution events (such as in an aircraft taxiing on the ground) inlets should be stoppered or back flushed to avoid strong contamination."

4) The effects occur between the location where ozone is added or present and somewhere in the detection chamber/system. During atmospheric measurements the reaction times are usually longer, as ozone enters the sampling line together with the sample air and then travel in common until the detection system. Please estimate the total reaction (travel) time in your laboratory system so that other instrument users can judge the problem in their configuration.

Thank you for this important suggestion. We have now included this information in the experimental section, as described below in answer to the question concerning lines 111f.

Minor concerns - General remark: I suggest to use the term "zero air" instead of "background air". The later usually characterizes "not polluted" sample air. Moreover, is your synthetic air really clean or VOC-free, so that its influence on your experiments and results can be excluded? –

Thanks. We replaced "background air" with "zero air".

The synthetic air was introduced from a commercial 50L gas cylinder. While there may be some unmeasured impurities in the synthetic air, they were considered negligible here as none of the monitored masses showed significant concentrations when measuring the synthetic air with PTR-ToF-MS and GC-MS. For the GC-MS, we should definitely see alkyl peaks in the synthetic air measurement, if it was contaminated with oil related products as sometimes occurs. This was not the case. Nevertheless, we cannot exclude some impurity on other masses, which may react to one of our target masses under ozone exposure.

 L. 73. Please add that ozone (as non-polar) molecule is little affected / solved in the water bubbler –

Thank you for this recommendation.  We added a sentence in the manuscript:

l. 77: "Thereafter, the air stream was led through ultra-pure water for humidification. Ozone as a non-polar molecule has a very low solubility in water and will therefore not be lost during the humidification process."

L. 78. Please use SI units, that is "hPa" instead of "mbar".

Thanks, we changed the unit to hPa.

L. 80. How H2O and RH was measured and where? What is your reference temperature for calculating RH, just the laboratory temperature and you assume that the temperature of the scrubber assembly is identical?

Thanks. RH was measured with a MSR145 datalogger, that includes a temperature sensor. We added a sentence in the experimental section (see below). The installation position in the setup is indicated with "RH" in Figure 1.

l. 110: "Additionally, the influence of 80 % relative humidity on the scrubber lifetime was tested. This RH level was chosen as an extreme to see whether or not it changes the scrubber performance. Relative humidity was measured with a humidity sensor that includes a temperature sensor (MSR145, MSR, Switzerland; indicated with "RH" in Figure 1)."

L. 111f (sections 2.2.2 and 2.2.3). Shortly describe the material (FEP, PFA, PEEK, silcosteel,…) and parts (sampling tubes, …) that are in contact with the sample air and on which surface reaction can occur. And what are the residence (reaction) times in your system? See also general comments. –

Thank you for this suggestion. We added the information

l. 129: "The PTR used a FEP inlet tubing (OD 1/4" (0.635 cm), inner diameter (ID) 1/8" (0.3175 cm)) with an inlet flow of 200~sccm. The distance between ozone scrubber and PTR housing was 1.85~m, resulting in an inlet residence time $t_{res}$ of ca. 4 s. In order to regulate the pressure in the drift tube during flight measurements, the sample air passes and adjustable O-ring (fluorinated propylene monomer (FPM) or nitrile butadiene rubber (NBR), $t_{res}$ ≤30 ms). The influence of the O-ring on VOC measurements was found to be zero without $O_3$ present, but has not been tested separately under ozone exposure. Inside the instrument, a 1 m line (ID 1 cm) made of polyetheretherketone (PEEK) is used (ca. 70 sccm, $t_{res}$ max. 1 s). Limits of detection (LOD) were <0.05~ppb, with a total uncertainty of 15-20~\%."

l. 146: "The system's inlet flow was 200 sccm, tubing length between GC inlet and ozone scrubber 2 m (OD 1/4" (0.635 cm), ID 1/8" (0.3175 cm)), which results in an inlet residence time of ca. 5 s. Inside the system, the sample air is exposed to silicosteel tubing (OD 1/16" (0.1588 cm), ID 0.02" (0.0508 cm), 40 sccm, $t_{res}$ < 1 s) and stainless-steel surfaces in the traps ($t_{res}$ 1.5 min). LODs were typically <0.03~ppb (acetaldehyde, acetone and acrolein <0.2~ppb) and the total measurement uncertainty approximately 10 %."

L. 112. "hPa"

Thanks, unit changed to hPa.

L. 117. "was" before 2.85 m

Thank you for finding this typo, we added the missing word.

L. 131f. A major topic of your paper is the influence of the ozone scrubber. To better understand its functioning, you should add more information in this section. Do you use a filter holder with an inner diameter of 37 mm (or less)? What is the air residence time in it? Is the scrubber just a "tissue filter" (prepared as you described) or more? –

Thank you for this remark. We added the following sentences:

l. 155: "In this study the scrubbers were prepared by soaking quartz fiber filters (37~mm, GE Healthcare Life Sciences, USA) in a 10 % (w/w) aqueous solution for 1 h followed by drying under a nitrogen flow of approximately 100 sccm at room temperature. This quartz filter was placed under a 47 mm PTFE-filter (Sartorius, Germany) in a Teflon filter holder. The smaller quartz filter was selected to avoid leaks at the filter holder (ID 47 mm, Reichelt Chemie Technik, Germany) previously

caused due to the thickness of the quartz filter. The volume of the filter housing is ca. 55 mL, resulting in a residence time of approximately 6 s with a flow rate of ~600 sccm."

L. 164f, Fig. 3. Why the enhancements for the GC and the PTRMS are so different?

Thank you for this question. We have added some text explaining this issue.

l. 203: "Both, the PTR- and GC-MS measured higher acetaldehyde mixing ratios

when O3 was above 150 ppb (see Figure 3). This indicates that the interference is not instrument specific but more likely a function of the common inlet tubing exposure to ozone. Note that the inlet lengths to GC and PTR were roughly the same. The higher enhancement of the GC acetaldehyde could be due to emission of oxidation products from the material of multiposition valves as described by Apel et al. In contrast to the PTR data, the ozone induced enhancement of the GC signal increases with acetaldehyde concentration. This effect can be due to the different materials used for the tubing inside the instruments: Deming et al. showed, that in glass and metal tubing competitive adsorption occurs, which depends on humidity, concentration and functionality of the analyte, while polymer tubing shows independent absorption. Our fast GC instrument is equipped with heated silico-steel tubing, which allows competitive adsorption, while the PTR is equipped with PFA tubing. Additionally, with increasing $O_3$ mixing ratios ozonolysis reactions during trapping are gaining importance. It seems, that the interferences on the VOC measurements caused by high ozone exposure are an effect of both, inlet line and instrument's surfaces."

L. 168/169. You write "This indicates that the interference is not instrument specific but more likely function of the common inlet tubing exposure to ozone.". However, independent on the filament problem of the GC (that is at 2 ppb), the enhancement between 7:00 and 8:00 (and 1000 ppb ozone) is a factor of ~3.5 higher for the GC. I would conclude that's not only an inlet tube effect or do you expect higher influences by the inlet sampling inside the GC. Please better explain this difference. - Fig. 4a. What is the reason of the missing signal drop of the MGC at ~3:30 (and the moderate drop of the PTR), when the cal (acetaldehyde) signal jumped to 0? As both signals directly turn to 0 when ozone is switched off, this appears like a reaction of ozone at the surfaces of the sampling lines. Would this also explain the last point, namely the different behavior of both instruments? –

Thank you for your comment. We agree that it is an effect of both, inlet line and instrument's surfaces (see response to the previous comment). We added some text to clarify this as described in the above comment. Additionally, we added the following in the discussion of Figure 4:

l. 215: "The acetaldehyde signal increased further between 400 and 1000 ppb $O_3$ to about 0.4 ppb (GC) to 0.5 ppb (PTR). Interestingly, the GC signal in Figure 4.b) did not drop when the standard gas level dropped to 0~ppb. This is an interesting observation that we currently cannot explain. No abnormal behavior in the GC-MS could be ascertained at this time including retention time shifts, tuning anomalies or changes in RH. We conclude that most likely it was an unlogged flow switching issue. Nevertheless, it does not interfere with our general observation, that the acetaldehyde signal is suffering positive interference under high ozone exposure, most likely due to ozonolysis reactions at the tubing surface."

L. 176: "have the same exact mass" ◊ "have exactly the same mass" –

Thank you, we included your suggestion.

 L. 178/179. a) what is the process for the signal increase if ozone is added to zero-air and b) does this process/effect explain the two issues further up? - Fig. 6. For what species "cal" stands for? … 0.5 ppb propanal and 0.5 ppb acetone, that is, in sum 1 ppb? Then both instruments would measure too little (Fig. 6a and the MGC for C4) or too high (PTRMS for C4). Please better explain this. –

a) Thanks. We addressed the issue as explained in the answer to your main comment 2) (as before):

l. 223: "The PTR-ToF-MS in $H_3O^+$ mode cannot separate the aldehyde from the ketone as they have exactly the same mass, i.e. the PTR-Tof-data presented here always shows the sum of propanal and acetone (C3), and butanal and MEK (C4) respectively and is therefore double the GC signals for the separated species. When measuring zero air, ozone increases the signal of the C3 and C4 carbonyls, starting at $O_3$ mixing ratios of 400 ppb, similar to acetaldehyde (cf. Figure 5). This is most likely due to the reaction of ozone with species attached/adsorbed at the walls of the instrument sampling systems or to unmeasured ozone reactive species in the zero air."

b) We double checked the raw data work-up for this experiment and found that there was indeed a problem for the PTR integration on this day (due to a software issue in the mass calibration). Therefore, we integrated again and requantified taking into account the dependence on the primary ions, drift tube pressure and temperature as before. For MGC we also found a problem in the background correction that resulted in slightly negative values for zero air. However, even with these corrections we cannot explain the low mixing ratios for the C4 carbonyls measured by MGC. We double checked the data, but did not find any indication for a drift in the standard gas concentration or the existence of a leak. However, those factors would influence all compounds present in the standard gas, not only butanal and MEK. If the reason was an instrument internal problem, it should be observed for all experiments, not

only on the day of the experiment presented in Figure 6b). As the low mixing ratios are measured under all $O_3$ conditions, a depletion through ozonolysis can also be excluded. The calibration on that day was linear. Usually, we performed a full calibration before and after each experiment. On that day however, the calibration in the morning failed, so we used only the calibration from after the experiment for the quantification. Generally, we expect a higher sensitivity during the first calibration after tuning (tuning was performed every morning) than after several hours of measurement. If this was the case and we use only the calibration data with less sensitivity i.e. lower area per ppb, we should a) measure too high concentrations during the experiment and b) see a drift towards higher concentrations with the measurement time. Neither is the case. In principal, new tuning between calibration and experiment could result in generally low mixing ratios, but we did only tune once in the morning (before the start of the experiment). Finally, we compared the response factor of the calibration from this day with the average response factor for all experiments. Unlike the C3 carbonyls, where the quantification is consistently correct and the response factor matches the average response factor for all experiments, the response factor for C4 carbonyls indicates less sensitivity during this day's calibration than on average. Again, this does not explain the low mixing ratios as low sensitivity during calibration (and corresponding higher sensitivity during the measurement) would result in too high measured mixing ratios during the experiment. It is certainly not satisfying that we could not find the reason for the C4 carbonyl quantification problem on that experimental day. However, we include the MGC data as the imperfect quantification does not interfere with our qualitative analysis of interferences and they fit to the qualitative results of the other experiments including the GC data. Additionally, the qualitative result for C4carbonyls is in line with the qualitative result of the measured C2-C3 carbonyls: The signals for the ketones (acetone (C3), MEK (C4)) increase with $O_3$ mixing ratios ≥400 ppb $O_3$ and the signals for the aldehydes (acetaldehyde (C2), propanal (C3) and butanal (C4)) are relatively stable with a tendency to decrease between 200 and 400 ppb $O_3$ and increase as well with $O_3$ mixing ratios ≥400 ppb.

We replaced Figures 4a) and 6 with the corrected data of both instruments:

[Figure]

*Figure 4a): Acetaldehyde mixing ratios at seven different ozone levels between 0 and 1000 ppb and 0 ppb standard gas level.*

*Figure 6: C3- and C4 carbonyl mixing ratios at approximately 0.5 ppb per VOC at different ozone levels.*

Additionally, we adjusted the text accordingly:

l. 231: „Interestingly, the aldehyde mixing ratios are relatively stable with a tendency to decrease with ozone when the standard gas was added. Figure 6 shows this phenomenon. Propanal and butanal mixing ratios do not show a substantial increase under the same $O_3$ conditions where they increase in the zero air measurement, while the sum of C3~carbonyls (PTR signal) and GC acetone again increase (as in the zero air measurement). As propanal slightly decreases and acetone strongly increases with ozone, the PTR measurements show a positive net ozone effect for the C3 carbonyls. For C4 carbonyls, the GC quantification during this experiment was compromised (too low mixing ratio) for unknown reasons. However, the qualitative results match the rest of our observations: butanal decreases slightly, while MEK increases slightly, leading to a stable signal for the sum of butanal and MEK, which is shown by the PTR data presented in Figure 6b. Additionally, the qualitative results of butanal and MEK are in line with the qualitative results of C2-C3 carbonyls: The signals for the ketones (acetone (C3), MEK (C4)) increase with $O_3$ mixing ratios ≥400 ppb and the signals for the aldehydes (acetaldehyde (C2), propanal (C3) and butanal (C4)) are relatively stable with a tendency to decrease between 200 and 400 ppb $O_3$ and increase as well with $O_3$ mixing ratios ≥400 ppb. Between 3.5 and 4 h after start of the experiment (cf. Figure 6), not all signals dropped to background levels. They finally drop once ozone was switched off. This is

consistent with the results from the zero air measurement (Figure 5) and the acetaldehyde data (Figure 4b)). It shows that exposure of the inlet tubing to high ozone does not rapidly clean the lines of the interfering compounds."

 L. 183. "Propanal and butanal mixing ratios decrease under the same O3 conditions …" not so clear in my opinion, because the m.r. appears to increase later in the time series. I would describe this with "indifferent with a tendency to depleted m.r. or so".

We agree with your observation and adjusted the text accordingly:

l. 231: „Interestingly, the aldehyde mixing ratios are relatively stable with a tendency to decrease with ozone when the standard gas was added. Figure 6 shows this phenomenon. Propanal and butanal mixing ratios do not show a substantial increase under the same $O_3$ conditions where they increase in the zero air measurement, while the sum of C3~carbonyls (PTR signal) and GC acetone again increase (as in the zero air measurement)."

 L. 190. "It shows that exposure of the inlet tubing to high ozone does not rapidly clean the lines of the artifact inducing compounds." That's an important sentence and I guess the first time where you describe what's going on. I suggest that you add a further short section 2.5 having a title like "Potential effects causing …" and a short description of a) surface reaction in the sampling lines, b) gas phase reactions in the sapling lines, c) gas phase reaction in the PTR detection system (e.g. in the drift tube or ??) and give them a real process name. Later in following sections you can refer to these three (or more?) "artefact reactions" and (if possible) detail the process further. –

Following the advice of the reviewer, we now added a short section 2.5 specifying "Potential effects causing interference".

l. 167: "VOC measurements performed by the PTR-ToF-MS and the fast GC-MS may in the presence of ozone, suffer interference through various effects. Surface reactions on the inner walls of the tubing can lead to ozonolysis of compounds previously absorbed on the FEP inlet tubing. The ozonolysis of alkenes, which are either present on the tubing surface or in the gas phase (sample air) can lead to production of carbonyl compounds which cause positive artifacts on the carbonyl masses. Another potential source of interference is fragmentation during the ionization process in the PTR-MS. Several groups reported for example fragments on PTR $m/z$ 69.07 from C5-C10 aldehydes (Buhr et al., 2002, Ruzsanyi et al., 2013, Wang et al., 2022). The instrument-internal fragmentation process itself is independent of ozone, but the presence of the aldehyde species in the sample air is likely to be caused by the release of those species from the sample line surface due to ozonolysis reaction. Not only the PTR, but also the GC-MS can suffer interference caused by ozone inside the instrument. It has been reported previously, that $O_3$

induced emission from rotor material of multiposition valves can lead to positive artifacts when measuring C2-C4 aldehydes (Apel et al., 2003)."

L. 215-217. This part I haven't understood fully. What you mean with "in the very beginning"? After using a new FEP line, after switching to a new air matrix, after …? And what you mean with "independent absorption"? Independent on what? –

Thanks, the first point is clarified as follows:

l. 269: "Furthermore, it can be seen that the sesquiterpene needs considerable time (more than an hour) to reach a steady state level in the beginning of the experiment (~03:00 in Figure 3), even with lines heated to 45 °C.

Thanks again for this point, we now describe this in section 3.1.2, as mentioned in the answer to your question concerning l.164:

l. 205: "In contrast to the PTR data, the ozone induced enhancement of the GC signal increases with acetaldehyde concentration. This effect can be due to the different materials used for the tubing inside the instruments: Deming et al. show, that in glass and metal tubing competitive adsorption occurs, which depends on humidity, concentration and functionality of the analyte, while polymer tubing shows independent absorption."

 L. 217-222. In my opinion, the text starting with "Note that …" fits better in the experimental section. –

Thanks for this suggestion. We moved the text to section 2.1 (l. 95).

l. 94: "The tests also included experiments without added VOCs to see if any of the analytes are produced in the pre-used inlet line. Note that during the very first experiment performed, flows were measured every time the VOC level was adjusted. It turned out that some compounds were emitted from the flow meter resulting in elevated terpene masses. When switching to a new calibration gas level as well as in the first hour of the experiment, there were spikes in the VOC signal. These were judged to be mechanical flow related anomalies and therefore removed to assure better visibility of the mixing ratio in the plots."

L. 227f (and Fig. 8). You say that the carbonyl compounds offset the expected signal drop with ozone or do you believe that the PTR doesn't show a depletion of the isoprene signal (in contrast to the GC) and you "only" see the positive offset from the carbonyl compounds? Please specify and clarify. –

Thank you for mentioning this. Of course, the PTR m69 signal would also drop if there was only isoprene present. We do believe that the positive offset caused by the

aldehydes is higher than the isoprene reduction and therefore we see an increasing PTR m69 signal. We added a sentence as clarification.

l. 280: "Interestingly the PTR-ToF shows slightly higher mixing ratios of the isoprene mass (*m/z* 69). The elevated signal on this mass can be caused by carbonyl compounds present in sample air or inlet line. Literature reports the same exact mass commonly used for isoprene detection in PTR systems to be a fragment of certain aldehydes (Buhr et al., 2002; Ruzsanyi et al., 2013). Most likely the PTR *m/z* 69 signal in the present study is elevated because under the experimental conditions the positive offset from the carbonyl compounds is higher than the isoprene depletion."

L. 248f (discussion of Fig. 10). Sorry, I can't follow your explanations. First, I couldn't figure out in Fig. 10 when or during which periods the scrubber is connected and when bypassed. Please add bars (or so) at the top. Furthermore, between ~5:00 ad ~7:20 at 2 ppb the sesquiterpenes never reaches the target concentration of 2 ppb, neither with nor without scrubber. I understand that this is due to the absorption/desorption (memory) effects. Still, it's hard to catch your main messages. It's a) that the scrubber has no influence (besides scrubbing ozone) and b) that the sesquiterpenes are strongly affected by memory effects? Is the memory effect limited to the sesquiterpenes and all other species just work fine and are a not affected by the ozone scrubber? Please improve your explanations. For a better understanding, it would help a lot adding times or time periods. –

On reflection we agree that this needs to be made clearer. The memory effect is limited to the sesquiterpenes, all other species work fine. For the conditions with 2ppb sesquiterpenes (Figure 10):

~4:30 (2 ppb sesquiterpenes, no Ozone, without scrubber), the sesquiterpenes reached steady state, but took more than 1.5h to reach it. At 5:00 (50 ppb O3, scrubber), they do not reach steady state. This can either be due to interaction with the scrubber or it can be due to reaction with the inlet line + filter holder on the "scrubber flow path" (cf. Figure 1: after the RH sensor, between the two three-port valves where the scrubber is connected). From 6:00 – 08:30, the sesquiterpenes do reach steady state in each condition, but the steady state mixing ratios are lower with higher ozone mixing ratios. With higher ozone mixing ratios, more sesquiterpenes will be depleted on their way to the filter scrubber, independent of possible reaction of the sesquiterpene with sodium thiosulfate.

To clarify this, we included specific time periods and improved the text as follows:

l. 306: Steady state is reached after almost two hours when the scrubber was introduced (~04:30). For the 30-40 min when the thiosulfate filters were introduced (~4:50-5:30) steady state is not even reached. Sesquiterpene adsorption by the

tubing material would again explain this effect. The scrubber was connected using Teflon tubing and a filter holder of the same material. The first time when the terpene-rich air was directed through the scrubber (50 ppb O₃, five hours after start of the experiment, (~4:50-5:30) the sesquiterpene mixing ratio increased within half an hour, while later (50 ppb & 170 ppb O₃, ~06:00-7:30) this was not the case. In other words, in the first half hour the sesquiterpenes appeared to be absorbed by the tubing (~4:50-5:30), while afterwards the material was conditioned and the concentration could reach steady state (~06:00-7:30). Steady state mixing ratios decrease with increasing O₃ mixing ratio as the scrubber was installed roughly half way between the junction where the standard gas was connected and the analytical instruments. On their way to the scrubber, the sesquiterpenes are depleted by ozonolysis. Furthermore, at the applied flow rate of approximately 650 sccm, there is already about 10 ppb of ozone passing the scrubber (cf. section 3.2) which allows the oxidant to further react with β-caryophyllene on the way between the scrubber and the detector. An improved filter assembly or several scrubbing filters as suggested by Pollmann et al. could improve the sesquiterpene sampling. For all compounds including isoprene no interference from the scrubber itself could be observed. Furthermore, interferences from ozone on the measurements of analytes like isoprene or acetaldehyde could be eliminated with the filter scrubber in line. The effect on sesquiterpenes could probably also be eliminated if the scrubber was placed at the inlet's front end and the flow through the scrubber was lower. However, it cannot be excluded that the sesquiterpene signal is affected by the scrubber material. Therefore, a longer time span with the scrubber in the sampling line would be required for the signal to reach steady state and the instrument would be unresponsive to rapid changes."

Additionally, we improved the plots by adding the ozone set value.

[Figure]

Figure 10. Terpene mixing ratios measured by PTR-MS with and without scrubber at 50 and 170 ppb O3.

L. 270. "Scrubber endurance" would be a more suited title. –

Thank you for this suggestion, we changed the title to "Scrubber endurance".

L. 287-290. These details on the filter assembly belong to section 2.3. I understand that you filter assembly has not been perfect, as you inserted the scrubber tissue into the existing somewhat larger Teflon filter, correct? Can you add here in this section or maybe better in the "lessons learned section" (see general comments) if – based on your experiences – the scrubber/filter assembly can be improved, e.g. by using a suitable filter housing (avoiding bypassing) and more important by adding more scrubber tissues? This will increase the scrubbing efficiency and the scrubber endurance (correct and do you expect a linear scaling with the number of scrubber tissues?), but do you also expect negative effects? –

Thank you for this remark. We moved the details on the filter assembly to section 2.3 and changed the sentence as following:

l. 341: "This 20 ppb offset during the second experiment might be caused by the filter not being perfectly centered in the filter holder. It is possible that a tiny stream of air bypassed the filter inside the filter holder as the 37 mm quartz filter was placed under a 47 mm Teflon as described in Section 2.3."

Additionally, we included some sentences in l. XY concerning several scrubber filters in a row and added a sentence within section 3.1.4 (Effect of sodium thiosulfate discussion on VOC measurements):

l. 379: "Exchanging the filter earlier does not affect the data quality and is therefore recommended. The filter assembly could be improved by using a filter housing which fits perfectly to the filter diameter in order to avoid any small air stream bypassing the filter. In this study, we did not examine the effect of several filters placed in series in the line as it was done by Pollmann et al. They found that with additional filters the scrubbing efficiency and scrubber endurance could be improved, but we adopted single filters to test their efficacy and endurance while minimizing potential uptake losses."

l. 317: "Furthermore, at the applied flow rate of approximately 650 sccm, there is already about 10 ppb of ozone passing the scrubber (cf. section 3.2) which allows the oxidant to further react with β-caryophyllene on the way between the scrubber and the detector. An improved filter assembly or several scrubbing filters as suggested by Pollmann et al. could improve the sesquiterpene sampling."

L. 346f. Only now/here (as you give a time period) I understand the cycle in Fig. 10. In Fig 15b, at 1:00 you switch to O3=50ppb and let this level until 2:00, but at 1:30 you add the scrubber and the measured O3 signal switch to 0, although the O3 level (by the O3 generator) still is at 50 ppb, correct? As requested before, please indicate this cycling in the relevant Figs better.

Yes, this is correct. We improved the figures by adding the O₃ set value.

[Figure]

Figure 15: Sesquiterpene mixing ratio with and without scrubber different O3 mixing ratios when 2 ppb of sesquiterpenes were introduced to the system; (a) 0 % RH, (b) 50 % RH.

---

## Author Comment (AC2)

**Reviewer 2**

Ernle et al. present an analysis that evaluates the role of ozone, and the impacts a understudied (and affordable) ozone scrubbing material, on VOC artifacts observed by PTR-MS and GC sampling. The authors first evaluate the role of ozone on producing artifacts in the absence and presence of VOCs, then evaluate the efficacy of implementing a $Na_2S_2O_3$ scrubber to remove ozone and limit inferred artifacts. The authors then characterize the scrubber determine important characteristics such as breakthrough, lifetime, and effects of humidity

Overall, the paper is well organized, easy to read, and the figures are easy to interpret. I agree with a number of the conclusions drawn by the authors regarding artifacts of aldehydes, but I have a number of concerns about the conclusions drawn from the observations of the alkenes. Specifically, I question whether these observations are an "artifact" – i.e., something artificially produced by the instrumentation and sampling setup - or a reflection of an ozonolysis experiment that is expected to occur when alkenes are mixed with ozone. My main comment (below) elaborates on this further, and I hope that the authors can dig deeper into the data to address this primary concern.

We thank the reviewer for the positive comments and for bringing up the discussion about the artifacts.

**Main Comment:**

My main comments pertain to the conclusions drawn from the various tests. What I haven't fully appreciated from the discussion is whether the tests really demonstrate an artifact, or simply shows that effect of VOC ozonolysis. To help frame my questions, I've listed the main takeaways I drew from the discussion and conclusions.

- In the presence of high ozone (but absence of VOCs), aldehyde and isoprene artifacts are observed due to ozone surface reactions with organics bound to Teflon tubing. Aldehydes are observed to increase in both the GC and PTR, while isoprene artifacts are observed by the PTR due to fragmentation of aldehydes.

  Yes, we agree.

- When steady mixing ratios of VOC standards are sampled by both instruments, the mixing of high ozone results in a positive aldehyde artifact. Part of this artifact results from surface reactions described in (1), while the remaining artifact may be due to reactions of VOCs with ozone in the tubing, instruments, or both.

  Yes, we agree.

- When steady mixing ratios of terpenes and sesquiterpenes are sampled by the instruments, the introduction of high ozone results in monoterpene and sesquiterpene decay for both instruments.

  Only for PTR, fast GC does not measure any terpenes with the method used here.

- The presence of the scrubber material removes ozone and limits decay of monoterpenes, but not sesquiterpenes.

  It prevents the decay of mono- & limits the decay of sesquiterpenes.

I'm convinced that conclusion (1) is consistent with a sampling artifact, and I fully agree with the authors that this presents an important consideration when sampling in high ozone environments.

Conclusions (2) and (3) are drawn from an experimental setup that essentially simulates VOC ozonolysis, and point (4) simply shows that removing ozone prevents VOC oxidation. So, is this really an artifact, or just an ozonolysis experiment? What I would like to know is if there is additional chemistry in the tubing (or in the instruments) that is prevented by the presence of the scrubber? Or in other words, if I don't have a scrubber, and I were to measure terpenes in the atmosphere, would I measure a bias because of surface reactions? Right now, as described, it is not clear to me that that this is true and if a scrubber is needed for preventing these potential artifacts

Thanks for these questions. The reviewer is correct, that in the presence of the scrubber gas-phase ozonolysis reactions are prevented (= ozonolysis experiment). The scrubber leads to a reduction of the gas-phase chemistry and therefore less interference. Additionally, the scrubber prevents ozone-surface reactions inside the tubing (and the GC), which cause for example the formation of positive acetaldehyde artifacts. If you don't have a scrubber and were to measure terpenes in the atmosphere, there would still be ozone induced depletion of the terpenes (gas phase), which could be prevented by placing the scrubber at the very front end of the inlet line. Unfortunately, we cannot be sure whether or not surface reactions are additionally occurring with the terpenes. An important point for us was, that the $Na_2S_2O_3$ scrubber does not interfere the terpene measurement, which means we can keep our list of target analytes as it is when using the scrubber.

I do think that the authors have the data to demonstrate whether these effects are present and perhaps can expand this discussion. For example, Fig. 3 seems to show that the GC has a higher positive bias than the PTR for acetaldehyde as the VOC mixture is increased at high ozone (i.e., the change in acetaldehyde from 50 – 1000 ppb is much higher for the GC than for the PTR). Is it possible that this is an artifact of the GC preconcentration? In such a case, I would agree that this an instrument artifact. Are similar observations made for the monoterpenes and sesquiterpenes?

I.e., are there relatively larger negative biases for the GC than for the PTR? Again, this would be convincing of a negative bias owing to instrument sampling and would warrant the use of a ozone scrubber to limit the sampling artifact.

Thank you for the question about the different acetaldehyde increase. We added some sentences for clarification on this point. We did not make similar observations for the terpenes with the GC, these were only measured by PTR.

l. 203: "Both, the PTR- and GC-MS measured higher acetaldehyde mixing ratios when O3 was above 150 ppb (see Figure 3). This indicates that the interference is not instrument specific but more likely a function of the common inlet tubing exposure to ozone. Note that the inlet lengths to GC and PTR were roughly the same. The higher enhancement of the GC acetaldehyde could be due to emission of oxidation products from the material of multiposition valves as described by Apel et al.. In contrast to the PTR data, the ozone induced enhancement of the GC signal increases with acetaldehyde concentration. This effect can be due to the different materials used for the tubing inside the instruments: Deming et al. showed, that in glass and metal tubing competitive adsorption occurs, which depends on humidity, concentration and functionality of the analyte, while polymer tubing shows independent absorption. Our fast GC instrument is equipped with heated silico-steel tubing, which allows competitive adsorption, while the PTR is equipped with PFA tubing. Additionally, with increasing $O_3$ mixing ratios ozonolysis reactions during trapping are gaining importance. It seems, that the interferences on the VOC measurements caused by high ozone exposure are an effect of both, inlet line and instrument's surfaces."

**Other comments**:

Line 64: Are there studies which show the effects of new vs. old tubing on VOC measurements?

To our knowledge there are no such studies, but Northway et al. mention that the history of the tubing is important. According to that study, the level of artifact production depends a) on the tubing (material, geometry) and b) on the tubing history. Conditioning with ozone improved the artifact formation, but was unfortunately a reversible effect.

Line 90: A lot of experiments were performed under a range of different VOC conditions. I might suggest including a table that lists out experiment conditions for clarity to the reader.

Thanks for this suggestion, we included a table with the different conditions at l. 73.

| Condition | O₃ levels / ppb | Calgas MR / ppb | RH / % |
|---|---|---|---|
| Effect of $O_3$ on VOCs | 0, 50, 1000 | 0, 0.5, 1, 2, 4 | 0 |
| Effect of $O_3$ on tubing | 0, 25, 50, 100, 150, 400, 750, 1000 | 0, 0.5 | 0 |
| Effect of RH on VOCs/scrubber | 0, 50, 150 | 0, 0.5, 2 | 0, 50, 80 |
| Scrubber endurance | 50, 150, 1000 | 0 | 0, 80 |

Line 114: While fragmentation is less of an issue for many of the analytes, a number of these species (and their products) do fragment (e.g. siloxanes, monoterpenes, sesquiterpenes, etc), and can impact important measurements of species such as isoprene, as demonstrated by the authors. I suggest rephrasing and referencing relevant fragmentation papers (e.g. Pagonis et al).

Thank you for this suggestion, we added some sentences to cover this point:

l. 125: "This is a soft ionization technique and therefore causes little fragmentation of the analytes during the detection process. This is the case for most analytes in this study. However, some species (e.g. terpenes, siloxanes) do fragment during ionization (Pagonis, 2019). Fragments can impact the measurement of target species such as isoprene if they have exactly the same mass."

Line 150 and Table 1: Could the authors propose a quantitative measure for each interference? For example, the increase in signal during zero VOC injection (e.g., amount of VOC signal produced per ppb of ozone introduced) would be helpful in quantifying positive artifacts owing to ozone interactions with the walls of the tubing and/or instrumentation.

We agree, that it would be helpful if there was a quantitative measure for each interference expressed in terms of ozone and tubing length. Unfortunately, according to Northway et al., the interferences from the tubing depend on both, tubing material and history of the tubing (previous exposure of the tubing to VOCs and ozone). This means that the specific degree of interference when unscrubbed high ozone levels are introduced, is dependent on the history of the inlet. Therefore, any relationship we derive is not generally applicable and might be misleading for the reader. Nevertheless, as an example we calculated the yield (ppb of VOC produced per ppb of ozone introduced) for acetaldehyde, which is the most affected compound measured by both instruments in our study. The average acetaldehyde yield for the GC $y_{acetaldehydeGC}$ was 0.0043±0.0055 per ppb of ozone while the average yield for the PTR $y_{acetaldehydePTR}$ was 0.0003±0.0011. The GC has higher production of VOC interference signal compared to the PTR. This is in line with our observations from Figure 3, that the GC signal increases with VOC and ozone concentration.

Table 1: As mentioned in my main comment, I'm still not quite sure if what is presented for monoterpenes and sesquiterpenes is an interference per say, and so I would be hesitant to include a down arrow for these species without further digging into the data and demonstrating that the high ozone is leading to additional biases beyond those of an ozonolysis experiment.

We agree, that the name 'artifact' might be misleading for compound depletion due to gas-phase ozonolysis reaction. Therefore, we replaced 'artifact' with 'interference' where necessary. Nevertheless, the aim of our study was to investigate the influence of O3 and humidity on VOC measurements. This includes also ozonolysis reactions, that happen during sampling. As Table 1 presents the effect of ozone on the measured species, we include ozonolysis reactions and sampling artifacts.

Figure 2: This caption feels incomplete and really doesn't describe what the authors are trying to show. It may be better to say "Effect of ozone mixing ratios (0, 50, 1000 ppb) on GC and PTR-MS measurements of chlorobenzene under a range of VOC mixing ratios ( X - X ppb)"

Thanks for spotting this. We rephrased as following:

"Figure 2: Effect of Ozone (0, 50, 1000 ppb) on GC and PTR-MS measurements of chlorobenzene under different standard gas levels (0-4 ppb)."

Lines 158 – 161: These sentences feel a bit distracting and I don't think are necessary for the discussion. This statement could be removed, or simply stated – e.g. "While the instruments observe slightly different mixing ratios owing to differences in sensitivity, the relative change in chlorobenzene mixing ratios remain unchanged when ozone is present."

Thank you for this suggestion, we changed the manuscript accordingly.

Line 165 - 166: Can the authors expand here? What trends or aspects of the measurements agree with what was observed by Northway et al. and Lehumpuhl et al.?

Thank you for the suggestion, we expanded the section accordingly.

l. 196: "Based on current literature, some carbonyl compounds such as aldehydes and ketones do have the potential to show higher mixing ratios when ozone is present. Northway et al. and Apel et. al observed positive artifacts under ozone presence for acetaldehyde. Additionally, Apel et al. observed artifact formation for propanal, acetone and butanal in their fast GC-MS system, which were emitted by parts of their system when ozone was present. Lehmpuhl et al. found positive interference also for larger carbonyl compounds."

Line 169 – 173: Here, the authors discuss the observations of acetaldehyde for each instrument independently. I'm also interested in the relative changes between the instruments. For example, it seems that the GC observes a significantly higher change in acetaldehyde at high O3 than the PTR. Is this due to a longer inlet line, differences in the residence time, or possibly differences in the instruments that that are causing these effects (e.g., ozone in the GC trap)? At first glance, this looks like this could be a combination of line impacts and instrument artifact for GC, while for the PTR this looks more like an effect of aldehydes production from the surface reactions of the tubing since the changes in the presence of ozone at the various VOC mixing ratios seem to match the changes in signal when VOCs are absent (as demonstrated in figure 4b).

Thank you for this comment, we agree with this interpretation. The inlet tubing lengths are kept the same for both instruments. Ozonolysis during trapping is likely. During field operation we use an internal ozone filter scrubber for the GC-MS, but for this experiment we removed this intentionally to investigate the influence of the scrubber on the VOC measurement. With increasing $O_3$ mixing ratios ozonolysis reactions during trapping become more important. Now, we discuss the differences in acetaldehyde enhancement more in detail, as already mentioned in the "Main comments" section.

l. 203: "Both, the PTR- and GC-MS measured higher acetaldehyde mixing ratios when O3 was above 150 ppb (see Figure 3). This indicates that the interference is not instrument specific but more likely a function of the common inlet tubing exposure to ozone. Note that the inlet lengths to GC and PTR were roughly the same. In contrast to the PTR data, the ozone induced enhancement of the GC signal increases with acetaldehyde concentration. This effect can be due to the different materials used for the tubing inside the instruments: Deming et al. show, that in glass and metal tubing competitive adsorption occurs, which depends on humidity, concentration and functionality of the analyte, while polymer tubing shows the phenomenon of independent absorption. Our fast GC instrument is equipped with silico-steel tubing, leading to competitive adsorption, while the PTR is equipped with PFA tubing. The higher enhancement of the GC acetaldehyde could also be due to emission of oxidation products from the material of multiposition valves as described by Apel et al. Additionally, with increasing $O_3$ mixing ratios ozonolysis reactions during trapping are gaining importance."

Line 178: Do the GC signals sum up to what is observed by the PTR?

Thanks for the question, we adjusted the text accordingly:

l. 226: "The PTR-ToF-MS in $H_3O^+$ mode cannot separate the aldehyde from the ketone as they have exactly the same mass, i.e. the PTR-Tof-data presented here always shows the sum of propanal and acetone (C3) and butanal and MEK (C4)

respectively and should be therefore double the GC signals for the separated species."

Since this was not the case for all experiments shown, we double checked the raw data work-up for this experiment and found that there was indeed a problem for the PTR integration on this day (due to a software issue in the mass calibration). Therefore, we integrated again and requantified, taking into account the dependence on the primary ions, drift tube pressure and temperature as before. For MGC we also found a problem in the background correction that resulted in slightly negative values for zero air. However, even with these corrections we cannot explain the low mixing ratios for the C4 carbonyls measured by MGC. We double checked the data, but did not find any indication for a drift in the standard gas concentration or the existence of a leak. However, those factors would influence all compounds present in the standard gas, not only butanal and MEK. If the reason was an instrument internal problem, it should be observed for all experiments, not only on the day of the experiment presented in Figure 6b). As the low mixing ratios are measured under all $O_3$ conditions, a depletion through ozonolysis can also be excluded. The calibration on that day was linear. Usually, we performed a full calibration before and after the experiment. On that day however, the calibration in the morning failed, so we used only the calibration from after the experiment for the quantification. Generally, we expect a higher sensitivity during the first calibration after tuning (tuning was performed every morning) than after several hours of measurement. If this was the case and we use only the calibration data with less sensitivity i.e. lower area per ppb, we should a) measure too high concentrations during the experiment and b) see a drift towards higher concentrations with the measurement time. Neither is the case. In principal, new tuning between calibration and experiment could result in generally low mixing ratios, but we did only tune once in the morning (before the start of the experiment). Finally, we compared the response factor of the calibration from this day with the average response factor for all experiments. Unlike the C3 carbonyls, where the quantification is correct and the response factor matches the average response factor for all experiments, the response factor for C4 carbonyls indicates less sensitivity during this day's calibration than on average. Again, this does not explain the low mixing ratios as low sensitivity during calibration (and corresponding higher sensitivity during the measurement) would result in too high measured mixing ratios during the experiment. It is not satisfying that we could not find the reason for the C4 carbonyl quantification problem on that experimental day. However, we include the MGC data as the imperfect quantification does not interfere with our qualitative analysis of interferences and they fit to the qualitative results of the other experiments including the GC data. Additionally, the qualitative result for C4carbonyls is in line with the qualitative result of the measured C2-C3 carbonyls: The signals for the ketones (acetone (C3), MEK (C4)) increase with $O_3$ mixing ratios ≥400 ppb $O_3$ and the signals for the aldehydes (acetaldehyde (C2), propanal (C3) and butanal (C4)) are relatively stable with a tendency to decrease between 200 and 400 ppb $O_3$ and increase as well with $O_3$ mixing ratios ≥400 ppb.

We replaced Figures 4a) and 6 with the corrected data of both instruments:

[Figure]

*Figure 4a): Acetaldehyde mixing ratios at seven different ozone levels between 0 and 1000 ppb and 0 ppb standard gas level.*

*Figure 6: C3- and C4 carbonyl mixing ratios at approximately 0.5 ppb per VOC at different ozone levels.*

Additionally, we adjusted the text accordingly:

l. 231: „Interestingly, the aldehyde mixing ratios are relatively stable with a tendency to decrease with ozone when the standard gas was added. Figure 6 shows this phenomenon. Propanal and butanal mixing ratios do not show a substantial increase under the same $O_3$ conditions where they increase in the zero air measurement, while the sum of C3~carbonyls (PTR signal) and GC acetone again increase (as in the zero air measurement). As propanal slightly decreases and acetone strongly increases with ozone, the PTR measurements show a positive net ozone effect for the C3 carbonyls. For C4 carbonyls, the GC quantification during this experiment was compromised (too low mixing ratio) for unknown reasons. However, the qualitative results match the rest of our observations: butanal decreases slightly, while MEK increases slightly, leading to a stable signal for the sum of butanal and MEK, which is shown by the PTR data presented in Figure 6b. Additionally, the qualitative results of butanal and MEK are in line with the qualitative results of C2-C3 carbonyls: The signals for the ketones (acetone (C3), MEK (C4)) increase with $O_3$ mixing ratios ≥400 ppb and the signals for the aldehydes (acetaldehyde (C2), propanal (C3) and butanal (C4)) are relatively stable with a tendency to decrease between 200 and 400 ppb $O_3$ and increase as well with $O_3$ mixing ratios ≥400 ppb. Between 3.5 and 4 h after start of the experiment (cf. Figure 6), not all signals dropped to background levels. They finally drop once ozone was switched off. This is consistent with the results from the zero air measurement (Figure 5) and the

acetaldehyde data (Figure 4b)). It shows that exposure of the inlet tubing to high ozone does not rapidly clean the lines of the interfering compounds."

Figure 10: Is this measurement by PTR or GC? Please specify in the caption.

We agree this would be helpful and adjusted the caption accordingly:

"Figure 10. Terpene mixing ratios measured by PTR-MS with and without scrubber at 50 and 170~ppb $O_3$."

Line 330: I presume humidity didn't have a strong effect on the oxygenates or other species that had a positive artifact during ozonolysis?

Your assumption is correct. We mention in line 389-391, that we did not observe any strong effects during this study due to drying/using a humid calibration:

l. 389: "For the measurements performed within this study, humidity did not have any influence on the GC- and PTR-ToF-MS instrument measurement capability as these dried the air before detection (GC-MS) or used humid calibrations (PTR-ToF-MS)."

Line 360: Citation for the stratospheric observations? Also, It would be helpful to compare these interferences to those observed in the stratospheric work to put into perspective the real-world implications of these interferences.

To address the reference request, we give the Apel et al. reference as the impetus to test their system for ozone was the stratospheric interference. We too have seen acetaldehyde anomalies in earlier PTR-MS data from the stratosphere, but as it was clearly erroneous it was not published.

As mentioned in the answer concerning your comment about line 150 and Table1, in our experiments the average acetaldehyde yield per ppb ozone for the GC $y_{acetaldehydeGC}$ was 0.0043±0.0055 while the average yield for the PTR $y_{acetaldehydePTR}$ was 0.0003±0.0011. The GC has higher production of VOC interference signal compared to the PTR. This is in line with our observations from Figure 3, that the GC signal increases with VOC and ozone concentration. The most obvious indicator of an ozone interference in stratospheric air was with acetaldehyde. For example, in one such incursion we saw similar dependencies: Our PTR measured 0.88 ppb acetaldehyde at an altitude of ~13000 m, when ozone was 465 ppb. Unfortunately, the fast GC-MS did not measure acetaldehyde during this campaign so a direct comparison cannot be made from this dataset.

We cover these points with the following added text.

l. 416: "Signals for acetaldehyde, propanal, acetone and butanal increased with ozone levels above 150 ppb in background measurements with GC as well as PTR.

Thus, it can be concluded, that there are positive artifacts generated in the experimental setup i.e. in the tubing, inside the ozone generator or within both of the mass spectrometers. As we observed the same during stratospheric measurements before, the ozone generator cannot be the only source. Apel et al. originally conducted ozone sensitivity tests on their airborne GC-MS system due to anomalous acetaldehyde observations in the stratosphere. Our experiments also show acetaldehyde to be the species most affected by ozone interference. When our PTR encountered a stratospheric intrusion in flight as on 2$^{nd}$ June 2020 we found 0.88 ppb acetaldehyde (altitude 13000 m, O$_3$ 465 ppb), extremely suspect for such a short-lived molecule under otherwise clean conditions. Unfortunately, the fast GC-MS did not measure acetaldehyde during this flight campaign. When 0.5 ppb of VOC standard gas was measured, the signals for propanal and butanal decreased due to the reaction with the OH radical, generated from ozonolysis of terpenes."

**Technical Comments:**

Line 27: Suggest "measured" in place of "covered"

We replaced the word:

l. 26: "With these measurement techniques a wide range of volatile organic compounds can be measured including aliphatic and aromatic hydrocarbons,[…]"

Line 31: Suggest "mixing ratios" in place of "values"

We replaced the word:

l. 33: "Northway et al. (2004) and Apel et al. (2003) reported for example increased mixing ratios for acetaldehyde in their systems for measurements in the lower stratosphere where ozone levels are high and humidity is low."

Line 62: It would be more helpful to point to Table 1 as opposed to Section 3 for the list of VOCs.

Thanks for the suggestion, we now refer to the table.

l. 61: "A list with the measured species is shown in Table 2."

Line 64: Suggest "instrument" or "technique" in place of "machine"

Thanks for the suggestion. We removed the word "machine".

l. 64: "Some species could be detected with both instruments simultaneously while other species could only be measured by one."

Line 95: Suggest "instruments" rather than "mass spectrometers" since the GC is not solely a mass spectrometer

l. 104: "For those experiments the flow through the scrubber was higher to provide enough air for both of the instruments."

Line 186: "Netto" should be "net"

Thanks for finding this typo. We corrected the word accordingly.

l. 235: "As propanal decreases and acetone increases with ozone, the PTR measurements shows the net ozone effect on C3 carbonyl."

Line 190: The wording is confusing - perhaps "compounds with positive artifacts"

Thanks. We adjusted the text as following:

l. 244: "It shows that exposure of the inlet tubing to high ozone does not rapidly clean the lines of the interfering compounds."

Line 261: Suggest "junction" in place of "T-piece"

Thanks for this suggestion. We replaced the word.